# Comparing PDSI drought assessments using the traditional offline approach with direct climate model outputs

Yuting Yang[1,7], Shulei Zhang[2,7], Michael L. Roderick[3,4], Tim R. McVicar[4,5], Dawen Yang[1], Wenbin Liu[6], Xiaoyan Li[2]

[1]State Key Laboratory of Hydroscience and Engineering, Department of Hydraulic Engineering, Tsinghua University, Beijing, China

[2]State Key Laboratory of Earth Surface Process and Resource Ecology, School of Natural Resources, Faculty of Geographical Science, Beijing Normal University, Beijing, China

[3]Research School of Earth Sciences, Australian National University, Canberra, ACT, Australia

[4]Australian Research Council Centre of Excellence for Climate Extremes, Canberra, ACT, Australia

[5]CSIRO Land and Water, Canberra, ACT, Australia

[6]Key Laboratory of Water Cycle and Related Land Surface Processes, Institute of Geographic Sciences and Natural Resources Research, Chinese Academy of Sciences, Beijing, China

[7]Equal contribution.

*Correspondence to*: Yuting Yang (yuting_yang@tsinghua.edu.cn)

**Abstract.** Anthropogenic warming has been projected to increase global drought for the 21st century when calculated using traditional offline drought indices. However, this contradicts observations of the overall global greening and little systematic change in runoff over the past few decades and climate projections of future greening with slight increases in global runoff for the coming century. This calls into question the drought projections based on traditional offline drought indices. Here we calculate a widely-used traditional drought index (i.e., the Palmer Drought Severity Index, PDSI) using direct outputs from 16 CMIP5 models (PDSI_CMIP5) such that the hydrologic consistency between PDSI_CMIP5 and CMIP5 models is maintained. We find that the PDSI_CMIP5-depicted drought increases (in terms of drought severity, frequency and extent) are much smaller than that reported when PDSI is calculated using the traditional offline approach that has been widely used in previous drought assessments under climate change. Further analyses indicate that the overestimation of PDSI drought increases reported previously using the traditional PDSI is primarily due to ignoring the vegetation response to elevated atmospheric $CO_2$ concentration ($[CO_2]$) in the offline calculations. Finally, we show that the overestimation of drought using the traditional PDSI approach can be minimized by accounting for the effect of $CO_2$ on evapotranspiration.

## 1 Introduction

Drought is an intermittent disturbance of the water cycle that has profound impacts on regional water resources, agriculture and other ecosystem services (Sherwood and Fu, 2014). By taking meteorological outputs from climate model projections as the inputs to offline drought indices/hydrological impact models, numerous studies have projected increases in future drought, in terms of severity, frequency and extent, mainly as a consequence of warming associated with anthropogenic climate change (Cook et al., 2014, 2015; Dai, 2011, 2012; Dai et al., 2018; Huang et al., 2015, 2017; Lehner et al., 2017; Liu et al., 2018; Naumann et al., 2018; Park et al., 2018; Samaniego et al., 2018; Sternberg, 2011; Trenberth et al., 2013). However, this substantial increase in projected drought contradicts observations of global vegetation greening and little systematic change in runoff over the past few decades and climate projections of future greening with slight increases in global runoff for the coming century (Alkama et

al., 2013; Greve et al., 2017; Labat et al., 2014; Roderick et al., 2015; Milly and Dunne et al., 2016; Scheff et al., 2017; Yang et al., 2018; Yang et al., 2019; Zhu et al., 2016). The scientific basis underpinning the projected drying trend using traditional offline drought indices/hydrological impact models is that the calculated increases in evapotranspiration ($E$) are larger than the projected increase in precipitation ($P$) in many places (Sternberg et al., 2011), which results in an increasing water deficit and thus increasing simulated future drought. However, direct climate model outputs of $E$ exhibit a much smaller increasing trend (Supplementary Figure S1) and the global land mean $P$ is actually projected to increase faster than its $E$ counterpart (Greve et al., 2017; Milly and Dunne, 2016, 2017; Roderick et al., 2015; Yang et al., 2018) leading to a very different conclusion.

Several recent studies have demonstrated that the drying bias in the offline calculated $E$ trend is primarily due to neglecting the impact of increasing atmospheric $CO_2$ concentration ($[CO_2]$) (and its resultant vapor pressure deficit increase) on the water use efficiency of vegetation (Lemordant et al., 2018; Milly and Dunne, 2016, 2017; Roderick et al., 2015; Swann et al., 2016; Yang et al., 2019). This vegetation-$[CO_2]$ response only impacts transpiration, not soil evaporation, interception from vegetation surfaces or sublimation in snow environments, however it should be noted that transpiration dominates (~ 65%; note that a transpiration over evapotranspiration ratio of $0.41 \pm 0.11$ are estimated by the CMIP5 models) global terrestrial evapotranspiration (Lian et al., 2018; Zhang et al 2016)). In existing hydrologic impact models/drought indices, $P$ and potential evapotranspiration ($E_P$; the rate of evapotranspiration that would occur with an unlimited supply of water) are the two key input variables, which respectively represent water supply to, and water demand from, the land surface. While $P$ is a direct climate model output, $E_P$ is not produced by climate models. The traditional approach is to calculate $E_P$ offline using the meteorological variables in the climate model output. The calculated $E_P$, together with the climate model projected $P$, are used to force an offline hydrologic impact model (or hydrologic calculations embedded in drought indices) that independently calculates $E$, runoff ($Q$), and storage change ($\Delta S$), for assessing hydrologic changes under future climate scenarios (the right-hand column shown in Figure 1). Among various $E_P$ models, the open-water-Penman model (Shuttleworth, 1993) and the reference crop Penman-Monteith model (Allen et al., 1998) have been most widely used in existing drought assessment studies, given their sound physical basis and relatively simple

formulations. Nevertheless, both Penman-based models do not faithfully capture the biological processes embedded in the climate models. The open-water-Penman model was designed for water surfaces, where surface resistance ($r_s$) is, by definition, equal to zero. Allen et al's (1998) reference crop Penman-Monteith model prescribed a constant $r_s$ at 70 s m$^{-1}$, which is appropriate for an idealized

reference crop in the current climate but does not account for the fact that $r_s$ increases with elevated [$CO_2$] over vegetated surfaces in climate model projections (Yang et al., 2019). As a result, existing traditional offline hydrologic impact models/drought indices calculate estimates of $E$, $Q$ and $\Delta S$ that are different from those same variables in the original fully-coupled climate model output. For that reason, the consequent assessments of drought changes in existing traditional offline hydrologic impact

models/drought indices do not correctly represent the projections in the underlying fully-coupled climate models. Figure 1 illustrates the inconsistency in the hydrologic predictions (also see Milly and Dunne 2017) that have resulted in different trends in projected future drought between climate models and traditional offline hydrologic impact models/drought indices.

Here, we re-assess changes in future global drought using climate model projections from 16 Coupled-
Model-Intercomparison-Project-Phase-5 (CMIP5) models under historical (1861-2005) and Representative Concentration Pathway 8.5 (RCP8.5; 2006-2100) experiments (Taylor et al., 2012). These 16 CMIP5 models were selected as they output all variables, including runoff, that are needed for the analysis performed herein. The Palmer Drought Severity Index (PDSI; Palmer, 1965) is adopted here to quantify drought as it has been widely used for operational drought monitoring and is

increasingly used in studies assessing drought under climate change (Cook et al., 2014, 2015; Dai, 2011, 2012; Dai et al., 2018; Lehner et al., 2017; Liu et al., 2018; Sheffield et al., 2012; Swann et al., 2016; Trenberth et al., 2013). To maintain consistency between the calculated PDSI and the CMIP5 models, we first calculate PDSI using direct hydrologic outputs (i.e., $P$, $E$, $Q$, $\Delta S$) from the CMIP5 models (PDSI_CMIP5; corresponds to the centre column in Figure 1; also see Methods). This procedure

provides a reference for the PDSI projections. We then replicate the traditional PDSI calculation by using only meteorological data as inputs to calculate the reference crop Penman-Monteith $E_P$ (PDSI_PM-RC) (the right-hand column shown in Figure 1). The inference is that this traditional offline approach that only responds to meteorological forcing will overestimate drought relative to the direct

climate model output because it does not consider the biological effect of elevated $[CO_2]$. To evaluate

that inference, we again re-calculate the PDSI using an offline formulation that considers both the same

meteorological forcing along with the biological effects of elevated $CO_2$ (Yang et al., 2019) (the left-

hand column in Figure 1).

## 2 Data and Methods

### 2.1 Climate model projections

We used outputs from 16 climate models participating in Phase 5 of the Coupled Model

Intercomparison Project (CMIP5; Supplementary Table S1) under historical (1861-2005) and RCP 8.5

(2006-2100) experiments (Taylor et al., 2012). We used monthly series of runoff, precipitation, soil

moisture, sensible and latent heat flux at the land surface along with near-surface air temperature, air

pressure, wind speed and specific humidity. All outputs from 16 CMIP5 models were resampled to a

common $1^{\circ}$ spatial resolution by using the first-order conservative remapping scheme (Jones, 1999).

### 2.2 Calculation of PDSI

The Palmer Drought Severity Index (PDSI) was used to quantify drought (Palmer, 1965). To minimize

the impact of initial conditions on PDSI estimates, the first 40 years (1861-1900) are used for model

spin-up with the analyses focused on the 1901-2100 period. Briefly, the PDSI model consists of two

parts: (i) a two-stage bucket model that calculates the monthly water balance components (i.e., $E$, $Q$ and

$\Delta S$) using $P$ and $E_P$ as inputs, and (ii) a dimensionless index that describes the moisture departure

between the actual precipitation and the precipitation needed to maintain a normal soil moisture level

for a given time (i.e., the climatically appropriate for existing conditions values; these values were

calculated for the entire period of 1901-2100). The soil available water capacity (AWC) needed for

PDSI calculation was derived from the Global Gridded Surfaces of Selected Soil Characteristics

(https://webmap.ornl.gov/ogcdown/dataset.jsp?ds_id=569). While this parameter is inevitably subject to

uncertainties, Sheffield et al (2012) demonstrated that the PDSI calculation is insensitive to AWC

inputs. Detailed descriptions of PDSI can be found in Palmer (1965). A drought event is identified with

negative PDSI values, with a more negative PDSI indicating a more severe drought, whereas moist
events are associated with positive PDSI values.

We calculated PDSI following Palmer (1965) yet calculated $E_P$ using the reference crop Penman-Monteith model (PDSI_PM-RC; the right-hand column in Figure 1). The Penman-Monteith model explicitly considers influences from both radiative and aerodynamic components and has been widely used in previous PDSI calculations (e.g., van der Schrier et al., 2011; Dai et al., 2011; Sheffield et al.,
2012). In addition, we also used a modified Penman-Monteith model (PM[$CO_2$]; detailed later in the Methods and also see Yang et al., 2019) that accounts for the impact of elevated [$CO_2$] on $r_s$ to calculate $E_P$ and then PDSI (PDSI_PM[$CO_2$]; the left-hand column in Figure 1).

Additionally, instead of using hydrological simulations from the simplified water balance model embedded in the original PDSI model, we also calculated PDSI by using direct hydrologic outputs $E$, $Q$,
$\Delta S$ from the 16 CMIP5 models (PDSI_CMIP5; the centre column in Figure 1). This approach ensures that PDSI_CMIP5 faithfully represented the CMIP5 output. As the original PDSI model depends on a two-stage ''bucket'' model of the soil, we correspondingly regarded the moisture in upper portion of soil column (integrated over the uppermost 10 cm) from CMIP5 models as the moisture in the first layer and the total soil moisture content as the available moisture in both layers (so differences between total
soil-depth representation in CMIP5 models may lead to differences in PDSI estimates from individual models but are unlikely to impact the PDSI changes). Moreover, since the estimation of the weighting factor that converts moisture anomalies into the PDSI index also requires knowledge of $E_P$, we used the $E_P$ computed from a modified Penman-Monteith equation that explicitly considers the biological effect of elevated [$CO_2$] (i.e., PM[$CO_2$]) (Yang et al., 2019). To comprehensively document how the different
PDSIs were calculated, we illustrate the calculation procedures of the different PDSIs in Figure 2. Additionally, Matlab codes with worked examples of the different PDSIs can be accessed through *https://github.com/zslthu/Calculate-PDSI-in-Matlab*. The PDSIs were calculated using outputs of each CMIP5 model in turn, and the ensemble PDSIs (averaging PDSIs over the 16 CMIP5 models) were used in the following analyses.

## 2.3 Calculation of Potential Evapotranspiration

Two potential evapotranspiration formulations were used to calculate $E_P$. The first is the reference crop Penman-Monteith $E_P$ model, which computes $E_P$ (mm day$^{-1}$) as (Allen et al., 1998):

$$E_P = \frac{0.408\Delta R_n^* + \gamma\dfrac{900}{T+273}uD}{\Delta + \gamma(1+0.34u)} \qquad (1)$$

where $\Delta$ (Pa K$^{-1}$) is the gradient of the saturation vapour pressure with respect to temperature, $\gamma$ (Pa K$^{-1}$) is the psychrometric constant, $R_n^*$ (MJ m$^{-2}$ day$^{-1}$) is the surface available radiation (i.e., net radiation minus ground heat flux), $D$ (Pa) is the vapour pressure deficit of the air at 2 m height, $u$ (m s$^{-1}$) is the wind speed at 2 m height. In the reference crop Penman-Monteith model, $r_s$ is prescribed as 70 s m$^{-1}$ and this parameter value is embedded into the equation.

In addition, we used a modified reference crop Penman-Monteith $E_P$ model (i.e., PM[$CO_2$]) that accounts for the impact of rising [$CO_2$] (expressed in ppm units) on $r_s$, as derived in Yang et al. (2019). The PM[$CO_2$] model calculates $E_P$ as:

$$E_P = \frac{0.408\Delta R_n^* + \gamma\dfrac{900}{T+273}uD}{\Delta + \gamma\{1+u[0.34+2.4\times10^{-4}([CO_2]-300)]\}} \qquad (2)$$

## 2.4 Determining the timing of global warming target

To demonstrate the impact of warming on drought changes, we assessed changes in PDSI_CMIP5 under two future warming targets: 1.5 °C and 2 °C warming above the pre-industrial level. The 1.5 °C and 2 °C warming levels have been extensively discussed (Huang et al., 2017; Lehner et al., 2017; Liu et al., 2018; Park et al., 2018; Samaniego et al., 2018), as they are the two key warming targets set in the Paris Agreement on climate change (UNFCCC, 2015). The timing when the global warming targets (i.e., $t_{1.5}$ and $t_2$) is reached in each of the 16 CMIP5 models was computed based on the model output of the near-surface air temperature ($T_a$). We first selected 1986-2005 as the baseline period, which is a

widely used reference period for climate impact assessment (Lehner et al., 2017; Liu et al., 2018; Park et al., 2018). Then, we applied a 20-year moving average filter to the global mean annual $T_a$ time series to remove the interannual fluctuations in annual $T_a$ (Liu et al., 2018; Park et al., 2018). Each 20-year moving average is indexed to its final year (for example, the 20-year running mean $T_a$ for 2080 is an average of $T_a$ for 2061–2080). Finally, $t_{1.5}$ and $t_2$ are respectively determined at the times when global mean $T_a$ reached 0.9 °C and 1.4 °C above the 1986–2005 baseline, as this period was at least 0.6 °C warmer than the pre-industrial level (Hawkins et al., 2017; Schleussner et al., 2016).

## 3 Results

### 3.1 Predicted drought changes

Figure 3 shows the global patterns of PDSI trends for the three PDSIs. Evident drought increases are depicted by PDSI_PM-RC across much of the North America, South America, central-to-south Europe, Congo Basin, southern Africa, southeast China and southern coastal areas of Australia (Figure 3a), as widely reported previously (Dai, 2011, 2012; Dai et al., 2018; Cook et al., 2014; Lehner et al., 2018; Liu et al., 2018). However, those broad scale trends are not identified by either PDSI_CMIP5 (Figure 3b) or PDSI_PM[$CO_2$] (Figure 3c). Compared with PDSI-PM-RC, both PDSI_CMIP5 and PDSI_PM[$CO_2$] show much smaller changes. This result clearly indicates an inconsistency between the PDSI_PM-RC that has been widely used in traditional offline calculations for drought assessment studies and the underlying CMIP5 models, as the PDSI_CMIP5 as used here is based on the direct hydrologic outputs ($E$, $Q$ and $\Delta S$) from CMIP5 models.

To examine changes in drought frequency and extent, changes in months under drought within each year and changes in land area subject to dry and moist extremes are respectively shown in Figures 4 and 5. In applications, a PDSI < -3.0 is considered to be severe drought conditions while a PDSI > 3.0 is considered exceptionally moist (e.g., Palmer, 1965; Liu et al., 2018). We find that months with PDSI_PM-RC < -3.0 increase substantially over areas where PDSI_PM-RC evidently decreases, suggesting an increased drought frequency in these regions (Figure 4a). However, when assessed with PDSI_CMIP5 and PDSI_PM[$CO_2$], these drought frequency increases largely diminish (Figure 4b and

4c). Yet, moving to PDSI_CMIP5 and PDSI_PM[$CO_2$] apparently do not reduce the widespread distribution of drought frequency increase compared to drought frequency decrease (Figure 4b and 4c, i.e., there are more land areas with increasing drought frequency than with decreasing drought frequency). Similar results are found for drought extent changes as severe drought during the 21[st] century increases by $0.2393 \pm 0.0942\%$ per year ($p<0.01$) for PDSI_PM-RC but only increases by $0.1099 \pm 0.0228\%$ per year ($p<0.01$) for PDSI_CMIP5 and $0.1178 \pm 0.0308\%$ per year ($p<0.01$) for PDSI_PM[$CO_2$], respectively (Figures 5a-c). By contrast, moist areas (i.e., PDSI > 3.0) are less divergent among the three different PDSIs, although the PDSI_PM-RC still shows the least wetting lands compared to the other two PDSIs (Figures 5a-c). Interestingly, both PDSI_CMIP5 and PDSI_PM[$CO_2$] depict the increase in drought area to be essentially equivalent as the increase in moist area (Figures 5a-c), which may suggest an overall unchanged PDSI_CMIP5 (PDSI_PM[$CO_2$]) series at the global scale (Supplementary Figure S2). The above results are largely retained when assessing changes at different thresholds (i.e., mild drought/moist events with PDSI < -1.0 and PDSI > 1.0, and moderate drought/moist events with PDSI < -2.0 and PDSI >2.0 (Figures 4d-4i and 5d-5i). The fact that the results based on PDSI_PM[$CO_2$] closely follow that of PDSI_CMIP5 highlights the importance of vegetation response to elevated [$CO_2$] in the control of future surface hydrological changes. This demonstrates the inconsistency between the PDSI_PM-RC and CMIP5 models is largely caused by ignoring the vegetation response to elevated [$CO_2$] in the PDSI_PM-RC calculations.

## 3.2 The effect of warming on drought changes

Warming has been identified as the key driver of the overall future drought increase in numerous previous studies (Cook et al., 2014, 2015; Dai, 2011, 2012; Dai et al., 2018; Huang et al., 2015, 2017; Lehner et al., 2017; Liu et al., 2018). To further understand the impact of warming on drought changes, we assessed changes in PDSI_CMIP5 at 1.5 °C and 2 °C warming above the pre-industrial level. The PDSI_PM-RC is also presented for comparison. Any substantial increase in drought is identified when PDSI for a future warming target decreased by 1.0 compared to PDSI during the 1986-2005 baseline (i.e., $\Delta$PDSI < -1). Additionally, only places where the $\Delta$PDSI < -1.0 threshold is reached in at least 8 CMIP5 model (out of the 16 CMIP5 models so 50% and more) are considered to be robust projections

and thus used herein. Based on the PDSI_CMIP5, our results show that almost nowhere on earth (only

0.06% of the global land area) is projected to have a substantial drought increase at the 1.5 °C warming

target, and this number only slightly increases to 0.77% at the 2 °C warming target (Figures 6a and b).

In comparison, substantial increase in drought is identified at 5.10 % and 13.41 % of the global land

area at the two warming targets, respectively, when PDSI_PM-RC is used (Figures 6a and c). More

places are projected to have a substantial drought increase under future warming if we relaxed the

threshold of PDSI change to -0.5 (i.e., $\Delta$PDSI < -0.5) (Figures 6d-f). Nevertheless, the PDSI_CMIP5

still shows a considerable smaller percentage of drying lands (6.2% and 10.0%) than the PDSI_PM-RC

(26.32% and 34.77%) under the two warming targets, respectively, particularly over North America,

much of Amazonia, Europe, the Congo basin and southeast China.

## 4 Discussion and concluding remarks

The above results clearly demonstrate an overestimation of drought severity, frequency and extent using

PDSI in many previous assessments of future drought (e.g., Cook et al., 2014, 2015; Dai, 2011, 2012;

Dai et al., 2018; Lehner et al., 2017; Liu et al., 2018). The overestimation is primarily caused by

neglecting the impact of elevated $[CO_2]$ on $r_s$ and consequently on $E_P$ in the traditional offline

calculation. As $E_P$ is neither used nor produced by climate models, an offline intermediate $E_P$ model is

required to estimate $E_P$ based on climate model outputs. However, conventional $E_P$ models, such as the

open-water Penman model and the reference crop Penman-Monteith model, involve an important

assumption that $r_s$ remains constant over time (Allen et al., 1998; Shuttleworth, 1993). This assumption

is in general valid for water surfaces and/or wet bare soils but is not valid over vegetated surfaces. Over

vegetated surfaces, on one hand, elevated $[CO_2]$ leads to a partial stomatal closure that increases $r_s$ (e.g.,

Ainsworth and Rogers, 2007) yet on the other hand, elevated $[CO_2]$ has "fertilized" vegetation resulting

in an increased foliage cover (e.g., Donohue et al., 2013; Zhu et al., 2016), which effectively suggests a

reduction in $r_s$. In addition, elevated $[CO_2]$ serves as the ultimate driver of climate warming in the

CMIP5 models and consequently leads to an increase in atmospheric vapor pressure deficit, which also

tends to increase $r_s$ (Lin et al., 2018; Novick et al., 2016).

While the net effect of elevated $[CO_2]$ on $r_s$ is still uncertain in the real world, a recent study clearly

showed that in CMIP5 models, elevated $[CO_2]$ increases $r_s$, which, with all else equal, results in a

decrease of $E_P$ and thus $E$ (Yang et al., 2019). Yang et al (2019) also showed that over vegetated

surfaces, an increase in $E_P$ caused by warming-induced vapor pressure deficit increase is almost entirely

offset by a decrease in $E_P$ caused by the increase in $r_s$ driven by elevated $[CO_2]$ in CMIP5 models. This

suggests that climate change does not necessarily lead to a higher $E_P$ over vegetated surfaces and hence

increased drought under $[CO_2]$ enrichment, which is consistent with CMIP5 model projections yet

contradicts the perception that "warming leads to drying" presented in many previous studies (Cook et

al., 2014, 2015; Dai, 2011, 2012; Dai et al., 2018; Huang et al., 2015, 2017; Lehner et al., 2018; Liu et

al., 2018; Park et al., 2018; Samaniego et al., 2018; Sternberg, 2011; Trenberth et al., 2013).

Additionally, it is worthwhile mentioning that the CMIP5 models do project topsoil moisture (within the

top 10 cm) declines with a very similar spatial pattern to changes in PDSI_PM-RC (Dai, 2012; Dai et

al., 2018), which might be important for wildfire risk and various biological processes that take place

close to the surface. However, since no systematic decline in runoff or in relevant vegetation parameters

(e.g., leaf area index and gross/net primary production) seems to result from it (Greve et al., 2017; Milly

and Dunne, 2016, 2017; Roderick et al., 2015; Swann et al., 2016; Yang et al., 2019), this decline in

topsoil moisture in the CMIP projections seems to have little influence from the vegetation and

hydrological perspectives. This is likely as root-zone or deeper soil moisture that is of more

agricultural/ecological and/or hydrological significance, is projected to remain more or less unchanged

(Berg et al., 2017; Greve et al., 2017), consistent with PDSI_CMIP5 and PDSI_PM$[CO_2]$ (Figure 3).

Here, we use PDSI as an illustrating case; but note that similar results were also found in another

commonly used drought index (i.e., the Standardized Precipitation-Evapotranspiration Index, or SPEI;

Vicente-Serrano, 2010) (Supplementary Figure S3). Nevertheless, both PDSI and SPEI, as well as other

drought/aridity metrics, are secondary offline impact models. Since climate models are fully-coupled

land (and ocean) – atmosphere models that are an internally consistent representation of the climate

system (Milly and Dunne, 2016), a scientific prior of applying any offline hydrological impact models

is that the adopted offline model must be able to recover the hydrological simulations generated by the

climate models (Roderick et al., 2015; Milly and Dunne, 2017; Yang et al., 2019). Otherwise, any

inconsistency in hydrological predictions between offline impact models and climate models themselves would lead to inconsistent predictions in other components of the climate system. Unfortunately, this important scientific prior has been largely ignored in many previous drought assessment studies, leading to biased drought predictions that are actually inconsistent with the climate model outputs.

In summary, we have shown that climate model projections of the global drought area under future climate change has been largely overestimated. Our results suggest that the "warming leads to drying" perception may be fundamentally flawed, primarily due to ignoring the vegetation response to elevated [$CO_2$] (also see Yang et al., 2019). However, despite a small overall trend globally, we find that both drying and wetting areas are simulated to increase towards the end of this century (Figures 5 and Supplementary Figure S4), suggesting an increased variability in surface hydrological conditions that will likely be associated with increased droughts and/or floods and reduced reliability of available water at local/regional scales (e.g., Kumar et al., 2014). In this light, attention should be paid to regions where droughts and/or floods are projected to most likely increase (e.g., Mediterranean Europe and Central America) and more efforts may be needed to mitigate the consequent impact there under climate change.

## Code availability

Matlab codes with worked examples of the different PDSIs can be accessed through *https://github.com/zslthu/Calculate-PDSI-in-Matlab*

## Data availability

The data that support the findings of this study are openly available (http://cmip-pcmdi.llnl.gov/cmip5/).

## Author contribution

Y. Yang and M. Roderick designed the study. S. Zhang and Y. Yang performed the calculation and drafted the manuscript. All authors contributed to results discussion and manuscript writing.

## Competing interests

The authors declare that they have no conflict of interest.

## Acknowledgements

This study was supported by the National Natural Science Foundation of China (Grant No. 41890821), the Qinghai Department of Science and Technology (Grant No. 2019-SF-A4), and the Ministry of Science and Technology of China (Grant No. 2019YFC1510604). M. Roderick ackonwledges the support of the Australian Research Council (CE170100023). T. McVicar acknowledges support from CSIRO Land and Water. We thank the HESS Editor and three reviewers for constructive comments that improved the study.

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

**List of figures**

**Figure 1: Conceptual plot illustrating the inconsistency in the hydrologic predictions between climate models and offline hydrologic impact models.** The symbols $P$, $E_P$, $E$, $Q$ and $\Delta S$ represent precipitation, potential evapotranspiration, actual evapotranspiration, runoff and storage change, respectively. The meteorological variables used to calculate $E_P$ depend on the adopted $E_P$ model but mainly include net radiation, near-surface air temperature, vapor pressure and wind speed. The biological factor here is the response of surface resistance to elevated $[CO_2]$ over vegetated lands.

**Figure 2**: **Flowchart of PDSI calculations.** Note that PDSI_PM-RC, PDSI_PM[$CO_2$] and PDSI_CMIP5 respectively follow the right-hand, left-hand, and centre columns in Figure 1.

**Figure 3: Global spatial pattern of PDSI trend. a-c**, spatial distribution of PDSI trends during 1901-2100 for (a) PDSI_PM-RC, (b) PDSI_CMIP5 and (c) PDSI_PM[$CO_2$], respectively. Black dots represent locations where the same sign of the PDSI trend is identified in at least 13 out of the 16 CMIP5 models (i.e., >80 % of models).

**Figure 4: Global spatial pattern of drought trends. a-c**, spatial distribution of trends in the number of months under severe drought (PDSI < -3.0) during 1901-2100 for (a) PDSI_PM-RC, (b) PDSI_CMIP5 and (c) PDSI_PM[$CO_2$], respectively. **d-f**, spatial distribution of trends in the number of months under moderate drought (PDSI < -2.0) during 1901-2100 for (d) PDSI_PM-RC, (e) PDSI_CMIP5 and (f) PDSI_PM[$CO_2$], respectively. **g-i**, spatial distribution of trends in number of months under mild drought (PDSI < -1.0) during 1901-2100 for (g) PDSI_PM-RC, (h) PDSI_CMIP5 and (i) PDSI_PM[$CO_2$], respectively.

**Figure 5: Time series of the global average fractional land area experiencing drought/moist conditions. a-c**, Global average time series of land area experiencing severe drought (PDSI < -3.0, red) and exceptionally moist (PDSI > 3.0, blue) conditions for (a) PDSI_PM-RC, (b) PDSI_CMIP5 and (c) PDSI_PM[$CO_2$], respectively. **d-f**, Global average time series of land area experiencing moderate drought (PDSI < -2.0, red) and moist (PDSI > 2.0, blue) conditions for (d) PDSI_PM-RC, (e) PDSI_CMIP5 and (f) PDSI_PM[$CO_2$], respectively. **g-i**, Global average time series of land area

experiencing mild drought (PDSI < -1.0, red) and moist (PDSI > 1.0, blue) conditions for (g) PDSI_PM-RC, (g) PDSI_CMIP5 and (i) PDSI_PM[$CO_2$], respectively. The solid curves represent the ensemble mean of 16 CMIP5 models and the shading represents the range by individual models. The time series are averaged over global land areas excluding Greenland and Antarctica.

**Figure 6: Areas with substantial drought increase under future warming**. **a,** Relative land area with substantial drought increase ($\Delta$PDSI < -1.0) under 1.5 $^{\circ}$C and 2 $^{\circ}$C warming based on PDSI_CMIP5 and PDSI_PM-RC. **b-c,** Spatial pattern of substantial drought increase ($\Delta$PDSI < -1.0) under 1.5 $^{\circ}$C and 2 $^{\circ}$C warming based on (b) PDSI_CMIP5 and (c) PDSI_PM-RC. **d-f,** Similar with a-c but for $\Delta$PDSI < -0.5.

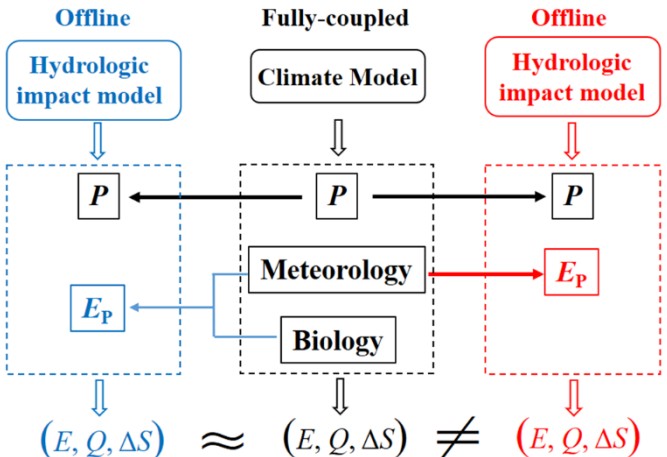

**Figure 1: Conceptual plot illustrating the inconsistency in the hydrologic predictions between climate models and offline hydrologic impact models.** The symbols $P$, $E_P$, $E$, $Q$ and $\Delta S$ represent precipitation, potential evapotranspiration, actual evapotranspiration, runoff and storage change, respectively. The meteorological variables used to calculate $E_P$ depend on the adopted $E_P$ model but mainly include net radiation, near-surface air temperature, vapor pressure and wind speed. The

biological factor here is the response of surface resistance to elevated [$CO_2$] over vegetated lands.

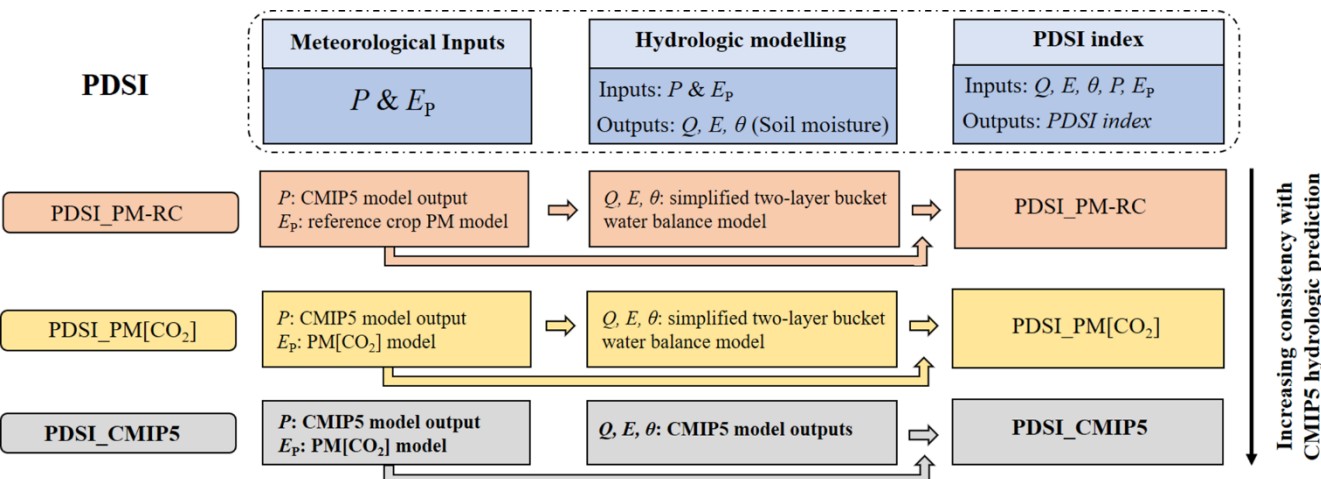

Figure 2: **Flowchart of PDSI calculations.** Note that PDSI_PM-RC, PDSI_PM[CO₂] and
PDSI_CMIP5 respectively follow the right-hand, left-hand, and centre columns in Figure 1.

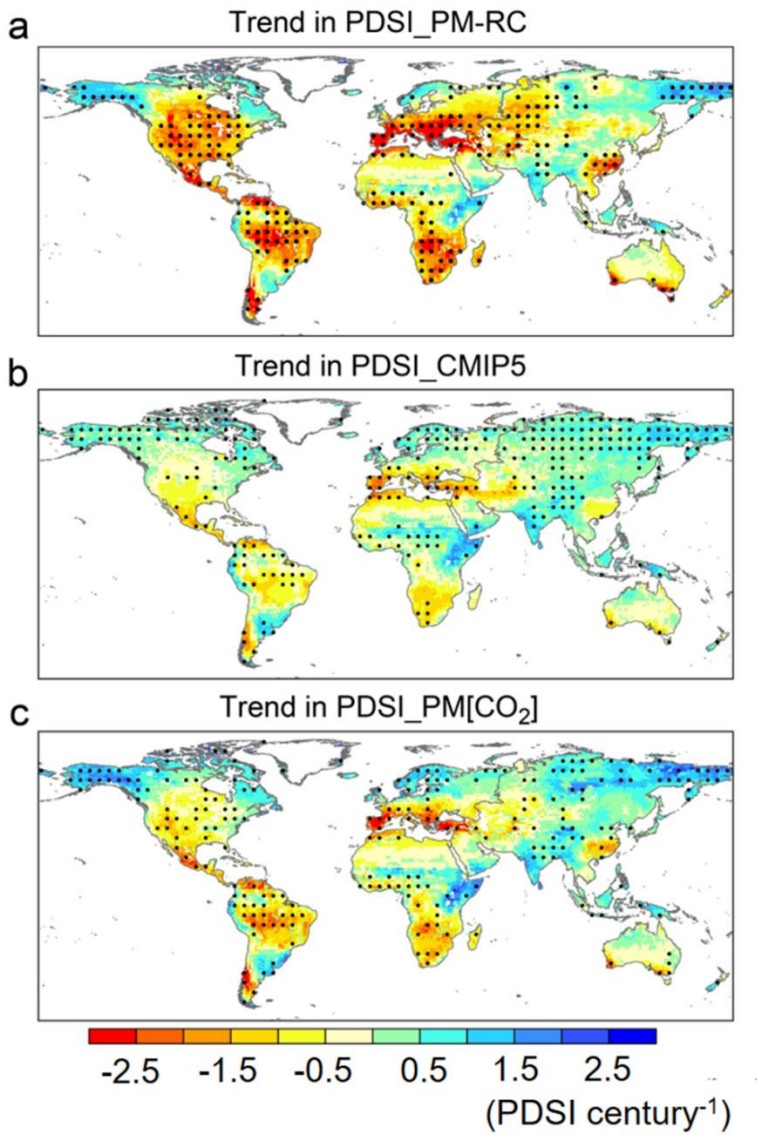

 **Figure 3: Global spatial pattern of PDSI trend. a-c**, spatial distribution of PDSI trends during 1901-2100 for (a) PDSI_PM-RC, (b) PDSI_CMIP5 and (c) PDSI_PM[$CO_2$], respectively. Black dots represent locations where the same sign of the PDSI trend is identified in at least 13 out of the 16 CMIP5 models (i.e., >80 % of models).

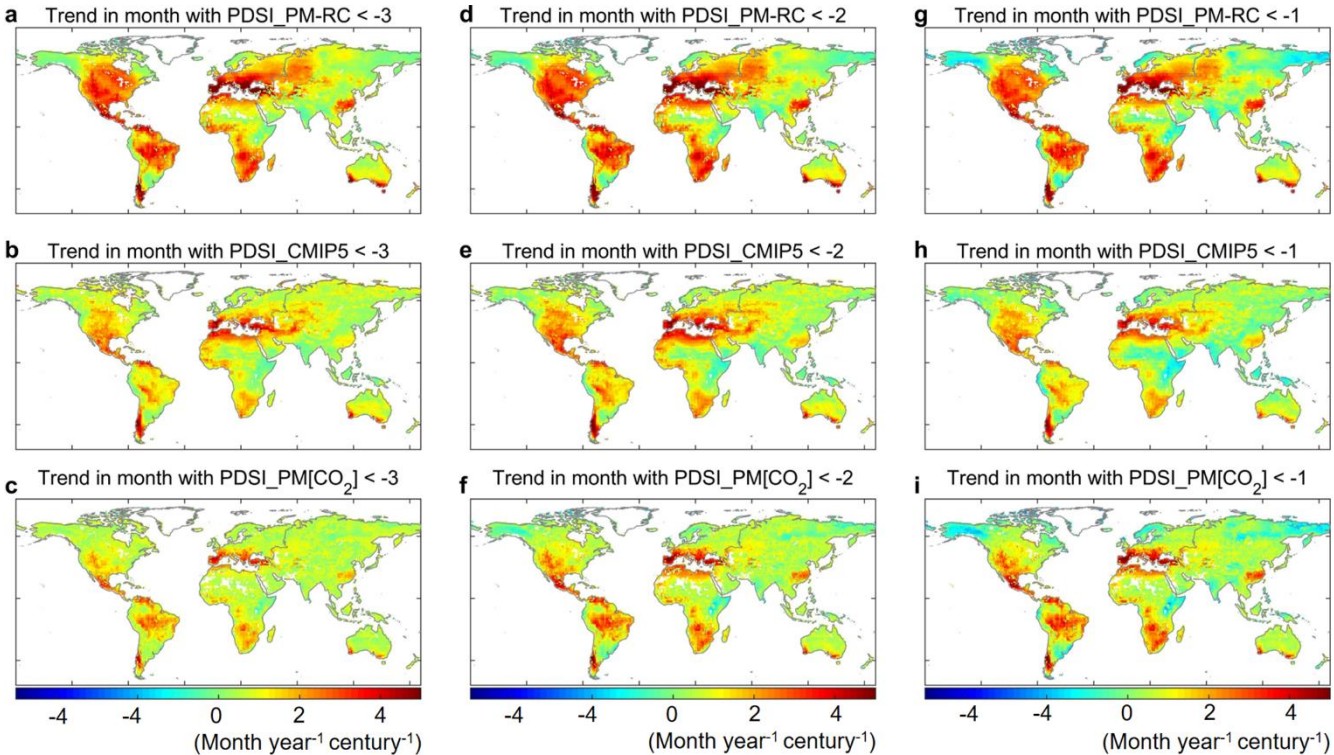

**Figure 4: Global spatial pattern of drought trends. a-c**, spatial distribution of trends in the number of months under severe drought (PDSI < -3.0) during 1901-2100 for (a) PDSI_PM-RC, (b) PDSI_CMIP5 and (c) PDSI_PM[$CO_2$], respectively. **d-f**, spatial distribution of trends in the number of months under

moderate drought (PDSI < -2.0) during 1901-2100 for (d) PDSI_PM-RC, (e) PDSI_CMIP5 and (f) PDSI_PM[$CO_2$], respectively. **g-i**, spatial distribution of trends in number of months under mild drought (PDSI < -1.0) during 1901-2100 for (g) PDSI_PM-RC, (h) PDSI_CMIP5 and (i) PDSI_PM[$CO_2$], respectively.

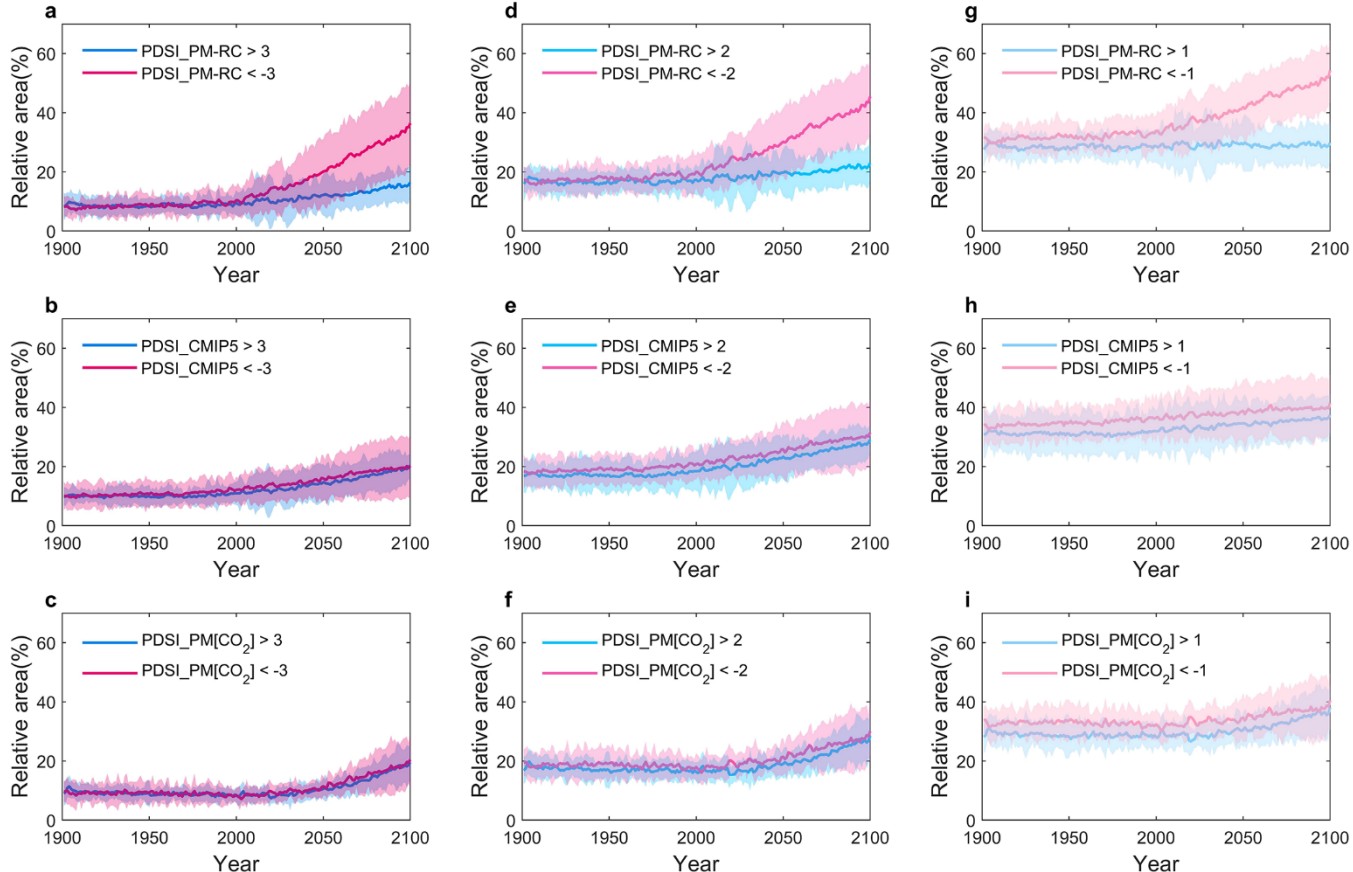

**Figure 5: Time series of the global average fractional land area experiencing drought/moist conditions. a-c**, Global average time series of land area experiencing severe drought (PDSI < -3.0, red) and exceptionally moist (PDSI > 3.0, blue) conditions for (a) PDSI_PM-RC, (b) PDSI_CMIP5 and (c) PDSI_PM[CO$_2$], respectively. **d-f**, Global average time series of land area experiencing moderate drought (PDSI < -2.0, red) and moist (PDSI > 2.0, blue) conditions for (d) PDSI_PM-RC, (e) PDSI_CMIP5 and (f) PDSI_PM[CO$_2$], respectively. **g-i**, Global average time series of land area experiencing mild drought (PDSI < -1.0, red) and moist (PDSI > 1.0, blue) conditions for (g) PDSI_PM-RC, (g) PDSI_CMIP5 and (i) PDSI_PM[CO$_2$], respectively. The solid curves represent the ensemble mean of 16 CMIP5 models and the shading represents the range by individual models. The time series are averaged over global land areas excluding Greenland and Antarctica.

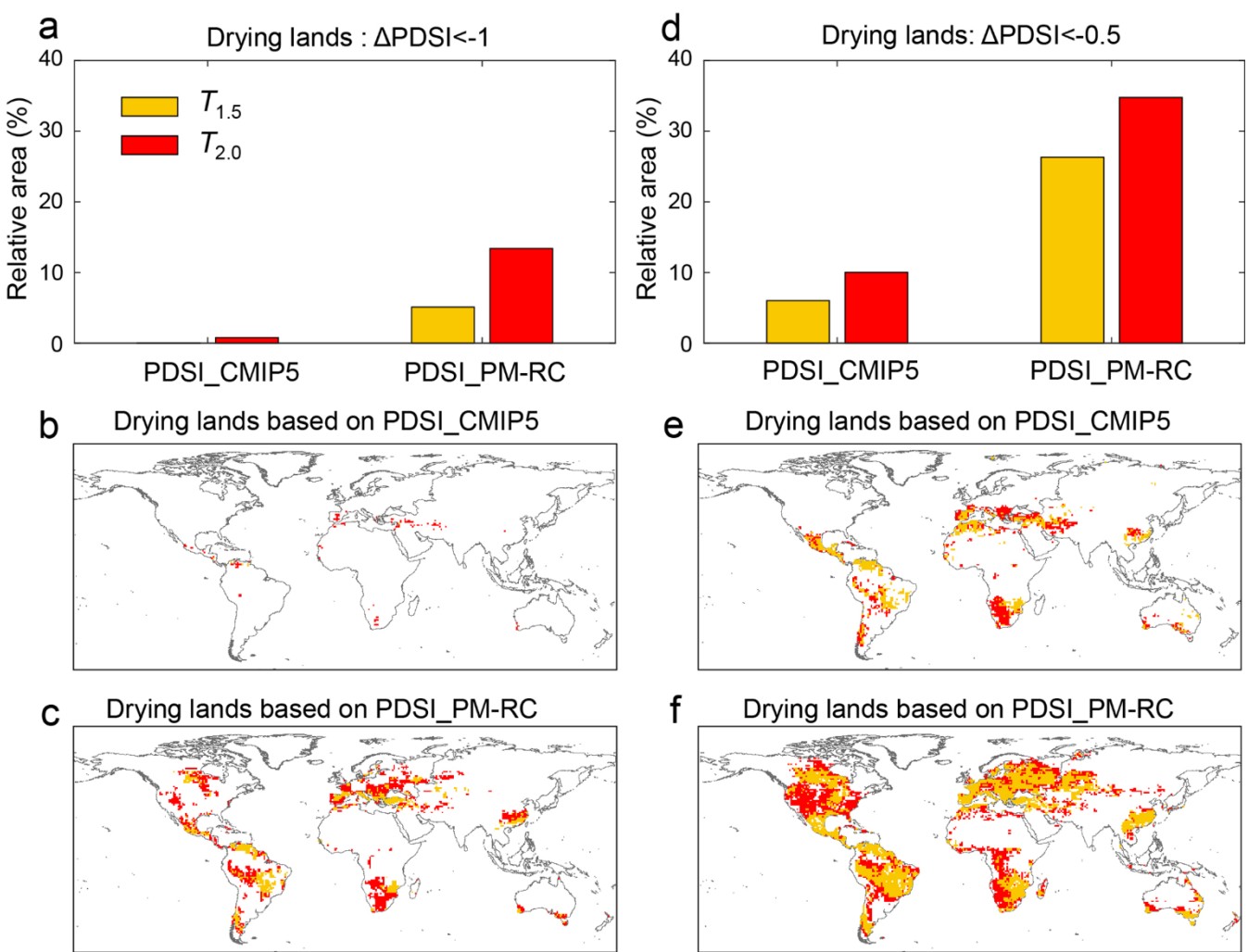

**Figure 6: Areas with substantial drought increase under future warming**. **a,** Relative land area with substantial drought increase ($\Delta PDSI < -1.0$) under 1.5 $^{\circ}$C and 2 $^{\circ}$C warming based on PDSI_CMIP5 and PDSI_PM-RC. **b-c,** Spatial pattern of substantial drought increase ($\Delta PDSI < -1.0$) under 1.5 $^{\circ}$C and 2 $^{\circ}$C warming based on (b) PDSI_CMIP5 and (c) PDSI_PM-RC. **d-f,** Similar with a-c but for $\Delta PDSI < -0.5$.