# Peer review of "Comparing PDSI drought assessments using the traditional offline approach with direct climate model outputs"

_Hydrology and Earth System Sciences, 2019_

## Short Comment (SC1) · 6 Feb 2020

Yang, Zhang and co-workers have presented an application of their $CO_2$-dependent modification of the Penman-Monteith equation to estimate ETP (Yang et al., 2019) for 16 CMIP5 climate projections (monthly until 2100). Based on the Palmer Drought Severity Index (PDSI) they work out the argument that inconsistencies arise, when only the hydrological variables are considered and adversary effects of higher $CO_2$ on stomatal conductance is neglected. They show that using the variables of the climate model predictions directly or their modified Penman-Monteith approach leads to consistent projections of no increase of global drought under climate change, which is very

different compared to PDSI calculations neglecting the effect of $CO_2$ on transpiration.

The presented study is highly relevant and covers one important aspect of current ecohydrological sciences under climate change. The manuscript is concise and well structured. It is transparently reporting the methods and results including the used matlab scripts. Since I have done a similar study recently (calculating the PDSI based on different climate model projections), I got interested in the study.

Despite all merits for the study and without claiming to have a practical solution to the problem, I have some concerns about the fundamental assumptions of the used approach.

**1 Only $CO_2$-effects reducing transpiration considered**

The authors consider two mechanisms: 1) that elevated atmospheric $CO_2$ directly reduces stomatal opening and 2) that higher $CO_2$ concentrations rises air temperature and leads to increased vapour pressure deficit and thus again stomatal closure. Hence both assumptions imply a reduced transpiration. Thus, the finding of their model might not be a result of competing mechanisms but of the assumptions and problem framing.

As much the authors argue for a more broad conscious about $CO_2$-effects, they neglect that stomatal conductance is not uniquely coupled with photosynthesis but also with cooling and other physiological processes. If plants could only reduce stomatal conductance, leaf temperatures would likely increase above operable levels. Urban et al. (2017) have shown such effects of stomatal opening under leaf temperature increase for cooling. They base their findings on measurements under controlled conditions separated from the effect of vapour pressure deficit on poplar and pine trees. Their results suggest that under stress photosynthesis and stomatal conductance become decoupled and thus transpiration could still increase with higher $CO_2$ and temperature.

**2 Penman-Monteith Equation**

Moreover, the Penman-Monteith Equation (which is fundamental to the study) has been criticised for limited capabilities to cover the actually claimed functionality (eg. Schymanski and Or, 2017) and to be consistent within the energy balance (eg. Kleidon and Renner, 2018). While from a practical point of view there is good reason to base studies on this equation, this cannot replace empirical evidence and/or detailed discussion of the implicit assumptions. Hence, the claim of the authors to be more correct with their "modified" model version without proper analysis appears a little weak.

**3 Palmer Drought Severity Index**

The PDSI calculates a very simple water balance – in the presented case with monthly time step. This implies a further hypothesis, which is about water availability to be evenly distributed over a month plus full water redistribution into the rhizosphere. Because water availability is another important control of stomatal conductance, the approach using PDSI on monthly data might overestimate water availability which would be in line with the reported findings?

**4 Conclusion**

There are many more aspects, which have to be and have been considered to predict responses of vegetation to elevated atmospheric $CO_2$ concentrations and temperature (which I have no doubt that the authors are aware of and partly participated in). Despite the freedom of the study to focus on one aspect alone, I find it difficult to allow for the main conclusion of the study based on the given situation of i) a model which

cannot account for trade-offs between different plausible effects, ii) very large scale and high level of aggregation, and iii) many implicit assumptions which have not been addressed.

I find it very helpful that the authors point out difficulties and traps of climate model output interpretations with respect to drought stress based on the PDSI and *offline* applications. In this respect, the manuscript makes a point, which is worth to be worked out. However, I do not see that the findings really *refute the common "warming leads to drying" perception*. Maybe a more detailed analysis and discussion of the Penman-Monteith model and measures to evaluate drought/wetness could be a way to substantiate the manuscript?

Despite all critics, I thank the authors for their work and the transparent presentation of their study. I think this is a good example how the open standards lead to higher quality and progress in our sciences.

All the best.

Conrad
* * *
**5  Bibliography**

Kleidon, A., and M. Renner (2018), Diurnal land surface energy balance partitioning estimated from the thermodynamic limit of a cold heat engine, Earth System Dynamics, 9(3), 1127–1140, doi:10.5194/esd-9-1127-2018.

Urban, J., M. W. Ingwers, M. A. McGuire, and R. O. Teskey (2017), Increase in leaf temperature opens stomata and decouples net photosynthesis from stomatal conductance in Pinus taeda and Populus deltoides x nigra, J. Exp. Bot., 68(7), 1757–1767, doi:10.1093/jxb/erx052.

Schymanski, S. J., and D. Or (2017), Leaf-scale experiments reveal an important omission in the Penman–Monteith equation, Hydrol. Earth Syst. Sci., 21(2), 685–706, doi:10.5194/hess-21-685-2017.

Yang, Y., M. L. Roderick, S. Zhang, T. R. McVicar, and R. J. Donohue (2019), Hydrologic implications of vegetation response to elevated $CO_2$ in climate projections, Nature Climate Change, 9(1), 44–48, doi:10.1038/s41558-018-0361-0.

---

## Referee Comment (RC1) · Anonymous Referee #1 · 24 Feb 2020

**General comments**

This paper analyzes projected changes in PDSI. It compares PDSI estimates obtained using potential evapotranspiration with and without accounting for the response of vegetation to increasing atmospheric CO2 concentration, as well as a direct estimate based on hydrological output from CMIP5 climate models. The main point is that there is no significant global drying trend based on PDSI, and the reason this was previously suggested is that offline impact models did not account for the response of vegetation to increased CO2. As noted by the authors (page 3, lines 54–57), several recent studies have already pointed out this problem when computing ET offline.

[Figure]

The valid point the authors make of refuting a general rule of "warming leads to drying" should not be interpreted as there will be no drying. The authors could try to make this even clearer by further emphasizing the projected increase in land area fraction under extreme conditions of water availability as well as the uncertainties in the projections.

Overall, the manuscript is well-structured and clearly conveys its main point. Nonetheless, it would be useful to further discuss some aspects of the methodology and address potential caveats of the PDSI.

Specific comments

1. Although PDSI has been a widely used index, it is not exempt from caveats. When analyzing projected changes in drought (water availability) it would be beneficial to also directly show the changes in relevant variables like soil moisture and precipitation minus evapotranspiration. Although results for SPEI are presented in the supplement, a summary of trends in projected soil moisture anomalies would be a suitable complement to the manuscript. Particularly, maps of the trends would provide a more comprehensive picture as opposed to the global averages.

2. It appears that the climatically appropriate for existing conditions (CAFEC) coefficients are estimated for the entire period 1901–2100 (if this is the case, it should be explicitly stated). This seems counterintuitive to me when analyzing projected changes. Why would it not be more meaningful to estimate the soil moisture anomalies relative to some reference conditions from the past or present, e.g. 1901–1960 as for SPEI in Fig. S4?

3. It would be relevant to discuss and/or provide sensitivity tests to assumptions underlying the calculation of PDSI. For example, what value was selected for the available water capacity (AWC)? Is it constant in space? Are the values model dependent?

4. It would be insightful to know more about the variability of PDSI given that all data is already available. For example, maps of change in the standard deviation of PDSI

from a future period relative to present-day can be shown in the supplement.

5. The manuscript concludes (page 11, lines 273–274) highlighting the increased spatial variability in surface hydrological conditions. In this context, it could be appropriate to also discuss local changes in temporal variability, see Kumar et al. (2013).

Kumar, S., Lawrence, D. M., Dirmeyer, P. A. & Sheffield, J. Less reliable water availability in the 21st century climate projections. Earth's Futur. 2, 152–160 (2013).

6. Page 6, lines 129–132: Potential for discussion. Differences between total soil-depth representation in CMIP models may lead to systematic differences in PDSI estimates from individual models.

7. It should be noted that the discussed response of vegetation to increasing CO2 applies to transpiration, but not to evaporation from the soil and canopy as well as snow sublimation. In this case, increasing CO2 and temperature would have a direct effect towards increasing evaporation.

8. Fig. 3 shows that even for direct CMIP5 output there can be a considerable increase in the land fraction experiencing extreme drought/moist conditions. These areas could be even larger if we were to consider the full spread of the CMIP5 ensemble as opposed to plus/minus one standard deviation. Is it reasonable to consider that differences in how individual models represent the response of vegetation to increasing CO2 could explain the spread in CMIP5 projections? This may be an important discussion point for the paper.

9. What is the reason why this particular subset of 16 CMIP5 models was used and not all models that are available?

10. Trends in vegetation greening are mentioned in the abstract. The following reference about hidden vegetation browning could be helpful.

Pan, N., Feng, X., Fu, B., Wang, S., Ji, F. and Pan, S.: Increasing global vegetation browning hidden in overall vegetation greening: Insights from time-varying trends, Remote Sens. Environ., 214, 59–72, doi:10.1016/J.RSE.2018.05.018, 2018.

Technical comments

1. In Palmer (1965), equation 26 appears to use monthly recharge (R) instead of long-term average R. This might be worth double checking since it seemed to me the average is used in the provided scripts.

2. Lines 45 and 225: Inconsistency in the reference Lehner et al., 2017 or 2018? There is only one reference entry.

3. Page 10, lines 23: I would delete the word "also" since the effects are opposite.

4. Page 11, line 273: Is Fig. 3b–f correct? Or Fig. 3b–c?

5. Figure 3: The selection criteria for where to have the black dots does not seem optimal. As it is now, it is showing all regions where the mean and median of PDSI have the same sign. I would suggest a different threshold for model agreement, e.g. black dots where at least 2/3 of the models agree in sign. Alternatively, it could be useful to include in the supplement maps of model agreement that are complementary to Figs. 3d–f.

6. Page 8, line 194 and 196: It should be Fig. 4.

---

## Author Comment (AC1) · 25 Feb 2020

Reply to interactive comment on "Little change in Palmer Drought Severity Index across global land under warming in climate projections" by Conrad Jackisch.

Comment (1): Yang, Zhang and co-workers have presented an application of their $CO_2$-dependent modification of the Penman-Monteith equation to estimate ETP (Yang et al., 2019) for 16 CMIP5 climate projections (monthly until 2100). Based on the Palmer Drought Severity Index (PDSI) they work out the argument that inconsistencies arise, when only the hydrological variables are considered and adversary effects of higher $CO_2$ on stomatal conductance is neglected. They show that using the variables

of the climate model predictions directly or their modified Penman-Monteith approach leads to consistent projections of no increase of global drought under climate change, which is very different compared to PDSI calculations neglecting the effect of CO2 on transpiration. The presented study is highly relevant and covers one important aspect of current ecohydrological sciences under climate change. The manuscript is concise and well structured. It is transparently reporting the methods and results including the used matlab scripts. Since I have done a similar study recently (calculating the PDSI based on different climate model projections), I got interested in the study. Despite all merits for the study and without claiming to have a practical solution to the problem, I have some concerns about the fundamental assumptions of the used approach.

Reply: Thanks for your encouraging comments and our reply to your specific comments are given below.

Comment (2): Only CO2-effects reducing transpiration considered. The authors consider two mechanisms: 1) that elevated atmospheric CO2 directly reduces stomatal opening and 2) that higher CO2 concentrations rises air temperature and leads to increased vapour pressure deficit and thus again stomatal closure. Hence both assumptions imply a reduced transpiration. Thus, the finding of their model might not be a result of competing mechanisms but of the assumptions and problem framing. As much the authors argue for a more broad conscious about CO2-effects, they neglect that stomatal conductance is not uniquely coupled with photosynthesis but also with cooling and other physiological processes. If plants could only reduce stomatal conductance, leaf temperatures would likely increase above operable levels. Urban et al. (2017) have shown such effects of stomatal opening under leaf temperature increase for cooling. They base their findings on measurements under controlled conditions separated from the effect of vapour pressure deficit on poplar and pine trees. Their results suggest that under stress photosynthesis and stomatal conductance become decoupled and thus transpiration could still increase with higher CO2 and temperature.

Reply: We absolutely agree with the additional mechanisms you mentioned. All

these effects and potentially others (which you did not mention in the above comment) are considered in our calculation of PDSI_CMIP5, since we use direct outputs of CMIP5 models to drive the PDSI calculation (so a fully coupled approach), if those mechanisms were also considered in the underlying CMIP5 models. The use of a CO2-corrected Penman-Monteith model and the corresponding PDSI (i.e., PDSI_PM[CO2]) is to demonstrate that previous overestimations of drought changes using an offline approach (i.e., PDSI_PM-RC ) is primarily caused by ignoring the vegetation response to elevated CO2, as demonstrated by a very close agreement between PDSI_PM[CO2] and PDSI_CMIP5 but a much more stronger drought increase by PDSI_PM-RC. It should be noted that there are still differences between PDSI_PM[CO2] and PDSI_CMIP5, suggesting that only accounting for the CO2 effect as we conducted here is not able to fully recover the coupled-processes in CMIP5 models. Nevertheless, the method we used to correct CO2 effect in the calculation of potential evapotranspiration in the Penman-Monteith model still provides an effective and simple (yet imperfect) way for offline assessment of hydrological changes in climate model projections.

Comment (3): Penman-Monteith Equation. Moreover, the Penman-Monteith Equation (which is fundamental to the study) has been criticised for limited capabilities to cover the actually claimed functionality (eg. Schymanski and Or, 2017) and to be consistent within the energy balance (eg. Kleidon and Renner, 2018). While from a practical point of view there is good reason to base studies on this equation, this cannot replace empirical evidence and/or detailed discussion of the implicit assumptions. Hence, the claim of the authors to be more correct with their "modified" model version without proper analysis appears a little weak.

Reply: We agree with this reviewer on the fundamental assumptions and limits of the Penman-Monteith model. However, the focus of this study is to demonstrate that previous estimates of PDSI using an offline approach without considering the CO2 effects deviates a lot from the underlying PDSI that can be calculated directly using CMIP5

model outputs of precipitation, evaporation, runoff and soil moisture. In that context, discussion and analysis of the limitations of the Penman-Monteith model are well beyond the scope of this study.

The claim of our "modified" model is more correct is well demonstrated by a close agreement between PDSI_PM[CO2] and PDSI_CMIP5, as well as in Yang et al. (2019, NCC, https://doi.org/10.1038/s41558-018-0361-0). In Yang et al. (2019), surface resistance is derived by inverting the Penman-Monteith model using direct outputs of actual evapotranspiration and other relevant meteorological variables, and they find that surface resistance (in the CMIP5 models) is very closely related with CO2. Yang et al. (2019) also acknowledged that the corrections they made is only direct relevant for analyzing CMIP5 model outputs (so is relevant in the current study), whether and to what extend the proposed correction could represent the real world remains an open question.

In short, a formal (and appropriate) assessment of PDSI changes in CMIP5 model should be based on PDSI_CMIP5 (as we highlighted in the manuscript, see Figure 5), and the correction of Penman-Monteith model by considering CO2 is one way that this can be done although we recommend the approach in PDSI_CMIP5.

Comment (4): Palmer Drought Severity Index. The PDSI calculates a very simple water balance – in the presented case with monthly time step. This implies a further hypothesis, which is about water availability to be evenly distributed over a month plus full water redistribution into the rhizosphere. Because water availability is another important control of stomatal conductance, the approach using PDSI on monthly data might overestimate water availability which would be in line with the reported findings?

Reply: The monthly time step is used because it is the most commonly time step used for PDSI calculation in a great many existing studies, and for long-term drought changes in other drought indices as well (e.g., SPEI).

Comment (5): Conclusion. There are many more aspects, which have to be and have

been considered to predict responses of vegetation to elevated atmospheric $CO_2$ concentrations and temperature (which I have no doubt that the authors are aware of and partly participated in). Despite the freedom of the study to focus on one aspect alone, I find it difficult to allow for the main conclusion of the study based on the given situation of i) a model which cannot account for trade-offs between different plausible effects, ii) very large scale and high level of aggregation, and iii) many implicit assumptions which have not been addressed. I find it very helpful that the authors point out difficulties and traps of climate model output interpretations with respect to drought stress based on the PDSI and offline applications. In this respect, the manuscript makes a point, which is worth to be worked out. However, I do not see that the findings really refute the common "warming leads to drying" perception. Maybe a more detailed analysis and discussion of the Penman-Monteith model and measures to evaluate drought/wetness could be a way to substantiate the manuscript?

Despite all critics, I thank the authors for their work and the transparent presentation of their study. I think this is a good example how the open standards lead to higher quality and progress in our sciences.

Reply: We well understand this reviewer's points and concern, and we readily acknowledge that CMIP5 models may not be complete. However, we again note that the PDSI_CMIP5 accounted for all the coupled processes that have been considered in the CMIP5 models. We also acknowledge that there may be large uncertainties in the CMIP5 models, but that is not an issue this study deals with. Here we are interested in obtaining an accurate representation of the model outputs. That is also the reason that our title includes a statement of ".... in climate projections". The conclusion of "warming does not lead to drying" is based on CMIP5 model projections: under widespread and persistent climate warming, some places show a drying trend and others show a wetting trend with little average changes on a global scale.

701, 2020.

---

## Referee Comment (RC2) · Anonymous Referee #2 · 3 Mar 2020

General Comments

This report is a welcome contribution to the ongoing discussion in the literature regarding how to characterize changes in drought incidence under the changing climate. The paper is a follow-up to the paper by Yang et al (2019), which presents an equation that generally captures the variation of effective stomatal resistance within CMIP5 models as a function of atmospheric carbon dioxide concentration. In this paper, that relation is used to show how a popular drought index—the PDSI—can be adapted to characterize drought in our world of greenhouse warming. A readily available and simple offline alternative to the usual PDSI (and, in particular, to the Allen et al. form of the

Penman-Monteith equation) will likely be of value to climate-change impacts analysts, many of whom may not be familiar enough with the biological processes in play or have the resources to model the processes with greater fidelity. That being said, it is important to evaluate the performance of the modified index carefully and to lay out clearly the assumptions and limitations in one place.

To some extent, the literature in this area has had a certain feel of X-vs-Y to it, X being increase of drought, and Y being no change in drought to speak of. This paper moves a bit toward the middle in acknowledging increases in drought incidence, but the overall presentation still has the feel of Y. Some specific suggestions for movement toward what might be a more balanced presentation are offered below for the authors' consideration.

Specific Comments

The title "Little Change. . ." (which echoes that of Sheffield et al.) places the paper in the Y category mentioned in the General Comments above. To me, and perhaps to other readers, "little" implies something along the lines of "nothing to worry about." The authors might consider modifying the title to avoid that implication.

The reference to "PDSI" in the title, without qualification, is potentially confusing. Other publications (as well as this one) have shown that the usual PDSI equation applied to climate-change projections do imply increased drought. Would it be appropriate to change "PDSI" to "Co2-aware PDSI" or something else that conveys that idea?

In general, the paper does a good job of citing the relevant literature. However, it's not clear to me that the abstract does justice to the previous literature (including the authors' own works) when it uses the phrase "resolve a paradox."

It's not immediately apparent what it means for the abstract to say that "global PDSI_CMIP5" remains generally unchanged. If this refers to the global average of the time average of the ensemble average of PDSI, then it is possible that the element

of variability in space, in time, and across models could be lost in translation. It's hard to think about "drought" without considering variability.

The statement in the abstract that "projected increase in PDSI drought reported previously is primarily due to ignoring the vegetation response" seems somewhat overstated when I look at Figure 3, which suggests that the increase is about 50% or so due to ignoring the biological response, leaving another 50% that is not due to that.

I did not carefully evaluate what was implied by lines 138-140: "The PDSIs were calculated using outputs of each CMIP5 model in turn, and the ensemble PDSIs (averaging PDSIs over models) were used in the following analyses," but that passage gave me pause. Won't averaging across models reduce both the temporal and spatial variability and thereby impact drought estimates?

lines 208-210. The criterion for substantial increase in drought appears to based on the change in the average value of PDSI rather than the change in the exceedance of a threshold. Is that a good measure?

lines 210-212. It was stated earlier that ensemble averages were used for the analysis. It's quite possible I've read through the paper too quickly; the authors might consider taking precautions to avoid letting the casual reader get confused.

lines 233-235. I am confused by "yet on the other hand" (which by the way sounds redundant in itself) combined with "also," since both effects are working in the same direction. The "also" seems out of place here. I get that "also" here was meant in the sense of "and here's another thing it does," but the current sentence structure doesn't work for me.

line 242-247. I think this is another place where the authors could relax away from the "Y" position mentioned in the general comments. It seems to me that the dryness near the surface might be important for wildfire risk and perhaps for various biological processes that take place close to the surface. This idea might even be allowed to

bubble up into the abstract.

Figures 1 and 2. The creation of these figure to convey what's going on is appreciated. Figure 2 takes a while to understand. It might help if the four black arrows and the plus signs were removed. It also might help if there were another column for how Ep is computed.

Figure 3. A map of trend in PDSI doesn't seem as useful as a map of trend in exceedance of some substantial value of PDSI.

lines 435-436. "where the same sign of the PDSI trend is identified in at least 8 out of the 16 CMIP5 models.." Taken alone, without additional explanation in the caption, it seems like this would be true everywhere.

Figure 4. This figure averages out a lot of information. Do the benefits of its inclusion outweigh the possible misunderstanding that it might generate? See also related comment above regarding "global PDSI_CMIP5" in abstract.

Figure 5. As mentioned elsewhere, change in (expected value of) PDSI might not be the best metric for change in drought. Change in exceedance of thresholds might be better. I wouldn't be surprised if these were quite parallel, but to leave that taken for granted could weaken the overall impact of the paper.

Technical Corrections

line 93. Delete "and"

line 233. Change "increase" to "increases"

line 270-271. Consider changing "due to the ignorance of" to "due to ignoring."

line 285. Change semicolon to period.

line 433. Change "e-f" to "d-f"

line 287. Add period.

---

## Referee Comment (RC3) · Anonymous Referee #3 · 9 Mar 2020

This very innovative and important study shows that when the familiar Palmer Drought Severity Index (PDSI) is computed directly from global Earth System Model output of precipitation, evaporation, runoff and soil moisture storage (rather than box-modeling all those quantities from an offline-computed potential evaporation of questionable accuracy as is traditionally done), the dire projections of ubiquitous future global drought from those traditional studies vanish. Instead, the PDSI projections become both wetting and drying depending on region, consistent with the *direct* simulations of runoff, deep-layer soil moisture, etc. by the ESMs but not with the traditional PDSI studies.

This is a key methodological advance and shows that the PDSI index itself is not flawed

under climate change, rather its known problems stem from the traditional potential-evaporation input which is inaccurate, leading to inaccurate inferred water flux changes. The inaccuracy of the traditional potential-evaporation input is because leading-order biological effects of changing $CO_2$ and vapor pressure are taken into account in the ESMs but not in the potential-evaporation calculation, as the authors show well here.

I recommend only minor revisions before publication, since I was anonymous Reviewer #2 on the earlier version of this study that was originally submitted to Environ. Res. Lett., and I already had my concerns largely addressed during that review process at that journal. My strongly recommended minor revisions are listed below.

30: This kind of parenthetical remark/qualification is appropriate for the body text, but I don't think is needed for the Abstract - it makes the Abstract too complicated and clunky. At least, that is how I read it. So I think you should either remove or greatly shorten this remark. You can put something like this in the body text instead.

54-57, 94-96, 122, 226, 271, 418: Should also mention the impact of increasing/elevated vapor pressure deficit, as you do in the Abstract. The direct effect of $CO_2$ is only part of the story, as you explain well at 235-237 but the text does not reflect here at all.

88-89: It should be clarified here that this corresponds to the center stream in Figure 1, parallel to how you point out the right stream and left stream later in the paragraph.

117-120: Similarly, this should mention that it is the right stream in Fig 1.

120-123: Similarly, this should mention that it is the left stream in Fig 1.

126-135: Similarly, this should mention that it is the center stream in Fig 1.

178: Should be 3d, not 3e.

238: As stated in previous review for Environ. Res. Lett., "our results" on this line will be read by most readers as meaning "the current study" (even though that's not what

[Figure]

you actually mean.) Since it's actually Yang et al (2019) that showed this key fact (not the current study), this needs to be rephrased to make that clear. It's largely the same authors, but different study, and the distinction is important.

Fig 2: Caption should point out which rows respectively correspond to the left, right and center streams of Fig 1.

Fig 3d-f: Stippling when >50% of models agree on sign of change is trivial - this will almost always be true (unless exactly 8 models have increase and exactly 8 have decrease.) Rather, you should stipple when, say, >67% or >80% of models agree on sign of change. This better filters out changes that are just noise.

It is true that I suggested 50% threshold in previous review, but that was for models with dPDSI < -1, not for basic sign of change!  50% makes sense if the criterion is dPDSI < -1 because that's not likely to occur by chance. But it doesn't make sense for dPDSI < 0 or dPDSI > 0, since that occurs most of the time by chance (unless *exactly* 8 models happen to have a decrease!)

Supp Fig S1: This is greatly appreciated, but I think it would have an even greater impact if you reversed the color scale in panels b and c (i.e. make negative green/blue, and positive yellow/brown.)  This is because in this context we are thinking of E as a loss term in the water budget, and so increasing trend of E -> more drying. (I know that in other contexts/purposes more E -> wetter, but here the purpose is clearly to indicate that panel c is not as "drying" as panel b, so the colors should intuitively reflect that!)

---

## Author Comment (AC2) · 16 Mar 2020

**Response to Reviewers' comments**

We greatly appreciate the reviewers providing valuable and constructive comments on our manuscript HESS-2019-701. We seriously considered each comment and will revised/improved the manuscript accordingly. The individual comments are replied below. In the following the reviewer comments are black font and our responses are blue and to assist with navigation we use codes, such as R1C2 (Reviewer 1 Comment 2).

Anonymous Referee #1

R1C1: General comments: This paper analyzes projected changes in PDSI. It compares PDSI estimates obtained using potential evapotranspiration with and without accounting for the response of vegetation to increasing atmospheric $CO_2$ concentration, as well as a direct estimate based on hydrological output from CMIP5 climate models. The main point is that there is no significant global drying trend based on PDSI, and the reason this was previously suggested is that offline impact models did not account for the response of vegetation to increased $CO_2$. As noted by the authors (page 3, lines 54–57), several recent studies have already pointed out this problem when computing ET offline.

The valid point the authors make of refuting a general rule of "warming leads to drying" should not be interpreted as there will be no drying. The authors could try to make this even clearer by further emphasizing the projected increase in land area fraction under extreme conditions of water availability as well as the uncertainties in the projections.

Overall, the manuscript is well-structured and clearly conveys its main point. Nonetheless, it would be useful to further discuss some aspects of the methodology and address potential caveats of the PDSI.

Reply: Thanks for your encouraging and constructive comments. Your individual comments are replied below.

Specific comments

R1C2: Although PDSI has been a widely used index, it is not exempt from caveats. When analyzing projected changes in drought (water availability) it would be beneficial to also directly show the changes in relevant variables like soil moisture and precipitation minus evapotranspiration. Although results for SPEI are presented in the supplement, a summary of trends in projected soil moisture anomalies would be a suitable complement to the manuscript. Particularly, maps of the trends would provide

a more comprehensive picture as opposed to the global averages.

Reply: The maps of trends in soil moisture and precipitation minus evapotranspiration have been shown in a few previous publications (e.g., Berg et al., 2017; Greve et al., 2017; Swann et al., 2016; Yang et al., 2019); we have cited these papers and summarized/discussed their findings in the manuscript.

An important motivation of this study is actually based on these previous findings that total soil moisture does not show notable changes and precipitation minus evapotranspiration (or runoff) shows a slightly increase but estimated drought increases substantially in the coming century. The current study is designed to solve this contradiction. Several studies have pointed out the issue of ignoring the $CO_2$ effect in offline ET (and/or runoff) estimations (as noted by the reviewer), with the findings have important implications on drought changes. This study goes one step further by directly focusing on drought, using a widely drought index – PDSI.

The spatial patterns of PDSI trend are shown in Figure 3. The global averaged PDSI series was intended to give an overall comparison between different PDSIs at the global scale (given comments by the editor and other reviewers, we will delete this global average PDSI series in the revised manuscript).

R1C3: It appears that the climatically appropriate for existing conditions (CAFEC) coefficients are estimated for the entire period 1901–2100 (if this is the case, it should be explicitly stated). This seems counterintuitive to me when analyzing projected changes. Why would it not be more meaningful to estimate the soil moisture anomalies relative to some reference conditions from the past or present, e.g. 1901–1960 as for SPEI in Fig. S4?

Reply: Both PDSI and SPEI are calculated for the entire period 1901-2100 (both indices calculate the monthly departure from climatological means, and the climatological means are computed as the mean over 1901-2100). With the calculated SPEI series, in Fig. S4, we show the long-term SPEI change relative to the 1901-1960 mean to better highlight the changes. We will make it clear in the revised manuscript.

R1C4: It would be relevant to discuss and/or provide sensitivity tests to assumptions underlying the calculation of PDSI. For example, what value was selected for the available water capacity (AWC)? Is it constant in space? Are the values model dependent?

Reply: We will explicitly state all relevant data/parameters used for PDSI calculation. The sensitivity of PDSI to AWC has been examined in a previous study (Sheffield et al., 2012), and the authors found that changes in AWC have only very minor impact

on PDSI estimates. We will discuss this point in the revised manuscript.

R1C5: It would be insightful to know more about the variability of PDSI given that all data is already available. For example, maps of change in the standard deviation of PDSI from a future period relative to present-day can be shown in the supplement.
Reply: Thanks for the suggestion. We will provide relevant results in the revised manuscript.

R1C6: The manuscript concludes (page 11, lines 273–274) highlighting the increased spatial variability in surface hydrological conditions. In this context, it could be appropriate to also discuss local changes in temporal variability, see Kumar et al. (2013). Kumar, S., Lawrence, D. M., Dirmeyer, P. A. & Sheffield, J. Less reliable water availability in the 21st century climate projections. Earth's Futur. 2, 152–160 (2013).
Reply: Thanks for the suggestion. We will discuss this point in the revised manuscript.

R1C7: Page 6, lines 129–132: Potential for discussion. Differences between total soil-depth representation in CMIP models may lead to systematic differences in PDSI estimates from individual models.
Reply: Reply: Thanks for the suggestion. We will discuss this point in the revised manuscript. In addition, we will update our results by showing mean and range (estimates from all individual models) instead of standard deviation in the revised manuscript. This better shows the difference between individual models.

R1C8: It should be noted that the discussed response of vegetation to increasing CO2 applies to transpiration, but not to evaporation from the soil and canopy as well as snow sublimation. In this case, increasing CO2 and temperature would have a direct effect towards increasing evaporation.
Reply: We agree with this point. We focus here the question: what would be the difference if we consider or ignore the vegetation response to elevated CO2. Increased CO2 will lead to warming through enhanced radiative forcing; this effect is reflected in the used meteorological variables (e.g., air temperature) as well as in the direct hydrological outputs of CMIP5 models (e.g., total ET). All those variables have been used in the calculation of PDSIs. We find that despite the increasing temperature (and likely increasing soil evaporation and snow sublimation), PDSI_CMIP5 only shows minor changes compared with PDSI_Penman that ignores the CO2 effect on transpiration.

R1C9: Fig. 3 shows that even for direct CMIP5 output there can be a considerable

increase in the land fraction experiencing extreme drought/moist conditions. These areas could be even larger if we were to consider the full spread of the CMIP5 ensemble as opposed to plus/minus one standard deviation. Is it reasonable to consider that differences in how individual models represent the response of vegetation to increasing CO2 could explain the spread in CMIP5 projections? This may be an important discussion point for the paper.

Reply: Thanks for the suggestion. We will discuss this point in the revised manuscript. In addition, we will update our results by showing mean and range (estimates from all individual models) instead of standard deviation in the revised manuscript to better show the difference between individual models.

R1C10: What is the reason why this particular subset of 16 CMIP5 models was used and not all models that are available?

Reply: These particular 16 CMIP5 models were used because these 16 models provide all outputs we need, in particular runoff estimations.

R1C11: Trends in vegetation greening are mentioned in the abstract. The following reference about hidden vegetation browning could be helpful. Pan, N., Feng, X., Fu, B., Wang, S., Ji, F. and Pan, S.: Increasing global vegetation browning hidden in overall vegetation greening: Insights from time-varying trends, Remote Sens. Environ., 214, 59–72, doi:10.1016/J.RSE.2018.05.018, 2018.

Reply: Thanks for the suggestion and we will mention this in the revised manuscript.

Technical comments

R1C12: In Palmer (1965), equation 26 appears to use monthly recharge (R) instead of long-term average R. This might be worth double checking since it seemed to me the average is used in the provided scripts.

Reply: The equation 26 in Palmer (1965) does not use monthly R but monthly climatological R: long-term mean R for each month. We follow that.

That equation is to estimate monthly weighting factors, so each month has only one weighting factor.

R1C13: Lines 45 and 225: Inconsistency in the reference Lehner et al., 2017 or 2018? There is only one reference entry.

Reply: Apology for the typo. We will correct it in the revised manuscript.

R1C14: Page 10, lines 23: I would delete the word "also" since the effects are opposite.

Reply: Thanks for your suggestion. We will update the text following your suggestion.

R1C15: Page 11, line 273: Is Fig. 3b–f correct? Or Fig. 3b–c?
Reply: Apology for the typo. We will correct it in the revised manuscript.

R1C16: Figure 3: The selection criteria for where to have the black dots does not seem optimal. As it is now, it is showing all regions where the mean and median of PDSI have the same sign. I would suggest a different threshold for model agreement, e.g. black dots where at least 2/3 of the models agree in sign. Alternatively, it could be useful to include in the supplement maps of model agreement that are complementary to Figs. 3d–f.
Reply: There is a typo in the caption of Figure 3. The black dots actually show the same sign detected in at least 13 models (so >80%). We will correct it in the revised manuscript.

R1C17: Page 8, line 194 and 196: It should be Fig. 4.
Reply: Apology for the typo. We will correct it in the revised manuscript.

References:
Berg et al., Divergent surface and total soil moisture projections under global warming, Geophysical Research Letters, 44, 236-244, 2017.

Greve et al., Simulated changes in aridity from the last glacial maximum to 4XCO2, Environmental Research Letter, 12, 114021, 2017.

Swann et al., Plant response to increasing CO2 reduce estimates of climate impacts on drought severity, PNAS, 113, 10019-10024, 2016.

Yang et al., Hydrological implications of vegetation response to elevated CO2 in climate projections, Nature Climate Change, 9, 44-48, 2019.

To Anonymous Referee #2:

R2C1: General Comments This report is a welcome contribution to the ongoing discussion in the literature regarding how to characterize changes in drought incidence under the changing climate. The paper is a follow-up to the paper by Yang et al (2019), which presents an equation that generally captures the variation of effective stomatal resistance within CMIP5 models as a function of atmospheric carbon dioxide concentration. In this paper, that relation is used to show how a popular drought index (the PDSI) can be adapted to characterize drought in our world of greenhouse warming. A readily available and simple offline alternative to the usual PDSI (and, in particular, to the Allen et al. form of the Penman-Monteith equation) will likely be of value to climate-change impacts analysts, many of whom may not be familiar enough with the biological processes in play or have the resources to model the processes with greater fidelity. That being said, it is important to evaluate the performance of the modified index carefully and to lay out clearly the assumptions and limitations in one place.

To some extent, the literature in this area has had a certain feel of X-vs-Y to it, X being increase of drought, and Y being no change in drought to speak of. This paper moves a bit toward the middle in acknowledging increases in drought incidence, but the overall presentation still has the feel of Y. Some specific suggestions for movement toward what might be a more balanced presentation are offered below for the authors' consideration.

Reply: Thanks for your encouraging and constructive comments. Your individual comments are replied below.

Specific Comments:

R2C2: The title "Little Change. . ." (which echoes that of Sheffield et al.) places the paper in the Y category mentioned in the General Comments above. To me, and perhaps to other readers, "little" implies something along the lines of "nothing to worry about." The authors might consider modifying the title to avoid that implication.

Reply: Thanks very much for the point. We will change the title to: Vegetation response to elevated $CO_2$ reduces drought increase under warming in climate projections.

To editor (Prof. Ryan Teuling): Given the suggestion from this reviewer, we have decided to change the title (shown above). As a result, to also follow your suggestion, we will remove the global mean PDSI series but focus our analyses on spatial patterns and areas/time under drought in the revised manuscript.

R2C3: The reference to "PDSI" in the title, without qualification, is potentially confusing. Other publications (as well as this one) have shown that the usual PDSI equation applied to climate-change projections do imply increased drought. Would it be appropriate to change "PDSI" to "Co2-aware PDSI" or something else that conveys that idea?

Reply: Please see our reply to R2C2.

R2C4: In general, the paper does a good job of citing the relevant literature. However, it's not clear to me that the abstract does justice to the previous literature (including the authors' own works) when it uses the phrase "resolve a paradox."

Reply: We will remove the phrase "resolve the paradox" in the revised manuscript.

R2C5: It's not immediately apparent what it means for the abstract to say that "global PDSI_CMIP5" remains generally unchanged. If this refers to the global average of the time average of the ensemble average of PDSI, then it is possible that the element of variability in space, in time, and across models could be lost in translation. It's hard to think about "drought" without considering variability.

Reply: We will rewrite the abstract to avoid any misunderstanding/misinterpretation of such.

R2C6: The statement in the abstract that "projected increase in PDSI drought reported previously is primarily due to ignoring the vegetation response" seems somewhat overstated when I look at Figure 3, which suggests that the increase is about 50% or so due to ignoring the biological response, leaving another 50% that is not due to that.

Reply: We will rewrite the abstract to avoid any misunderstanding/misinterpretation of such.

R2C7: I did not carefully evaluate what was implied by lines 138-140: "The PDSIs were calculated using outputs of each CMIP5 model in turn, and the ensemble PDSIs (averaging PDSIs over models) were used in the following analyses," but that passage gave me pause. Won't averaging across models reduce both the temporal and spatial variability and thereby impact drought estimates?

Reply: We will update our results of Figure 3 to also show the range of drought/moist areas projected by all individual models. In addition, results in Figure 5 are not ensembles but the results agreed in at least 8 models.

R2C8: lines 208-210. The criterion for substantial increase in drought appears to based on the change in the average value of PDSI rather than the change in the

exceedance of a threshold. Is that a good measure?

Reply: The focus here was the increase in drought (decrease in PDSI by definition). A place changes from extreme moist to mild moist is also considered as an indication of potential drought increase. The same analysis has been applied in a few previous studies (e.g., Liu et al., 2018) so we use the same approach to be able to compare with results from others.

Changes of PDSI exceeding a certain threshold indicate changes area under drought, and this results in given in Figure 3.

R2C9: lines 210-212. It was stated earlier that ensemble averages were used for the analysis. It's quite possible I've read through the paper too quickly; the authors might consider taking precautions to avoid letting the casual reader get confused.

Reply: This is not the ensemble result; instead, it is the result that shared by at least 8 models.

R2C10: lines 233-235. I am confused by "yet on the other hand" (which by the way sounds redundant in itself) combined with "also," since both effects are working in the same direction. The "also" seems out of place here. I get that "also" here was meant in the sense of "and here's another thing it does," but the current sentence structure doesn't work for me.

Reply: We will restructure the sentence to avoid any misunderstandings.

R2C11: line 242-247. I think this is another place where the authors could relax away from the "Y" position mentioned in the general comments. It seems to me that the dryness near the surface might be important for wildfire risk and perhaps for various biological processes that take place close to the surface. This idea might even be allowed to bubble up into the abstract.

Reply: Thanks for your suggestion. We will extend the discussion on the possible implications of surface soil moisture change and will consider to also mention that in the abstract.

R2C12: Figures 1 and 2. The creation of these figure to convey what's going on is appreciated. Figure 2 takes a while to understand. It might help if the four black arrows and the plus signs were removed. It also might help if there were another column for how Ep is computed.

Reply: We will remove the four black arrows and the plus signs in the revised manuscript. The second column (under Meteorological Inputs) shows how $E_P$ was computed for each PDSI.

R2C13: Figure 3. A map of trend in PDSI doesn't seem as useful as a map of trend in exceedance of some substantial value of PDSI.

Reply: The map of PDSI trend gives a general information on how PDSI changes. We also show trend of area under drought (PDSI exceeds a certain threshold). Following your suggestion, we will add a map showing time (i.e., the number of months) in each year with a PDSI value exceeds a certain threshold.

R2C14: lines 435-436. "where the same sign of the PDSI trend is identified in at least 8 out of the 16 CMIP5 models.." Taken alone, without additional explanation in the caption, it seems like this would be true everywhere.

Reply: There is a typo in the caption of Figure 3. The black dots actually show the same sign detected in at least 13 models (so >80%). We will correct it in the revised manuscript.

R2C15: Figure 4. This figure averages out a lot of information. Do the benefits of its inclusion outweigh the possible misunderstanding that it might generate? See also related comment above regarding "global PDSI_CMIP5" in abstract.

Reply: As also suggested by the editor, we will remove this figure in the revised manuscript.

R2C16: Figure 5. As mentioned elsewhere, change in (expected value of) PDSI might not be the best metric for change in drought. Change in exceedance of thresholds might be better. I wouldn't be surprised if these were quite parallel, but to leave that taken for granted could weaken the overall impact of the paper.

Reply: Please see our reply to R2C8.

In addition, we understand this reviewer's concern of using changes in PDSI as the measure. However, using changes in exceedance of thresholds may also incur other issues. For example, if we chose PDSI = -1 as the threshold for drought, then locations having a PDSI = -2 in the baseline period and a PDIS = -1.5 in the future period (or PDSI = -1.5 in the baseline period and PDSI = -1.51 in the future period) will be identified as places with a substantial drought increase.

Technical Corrections
R2C17: line 93. Delete "and"
Reply: Will revise in the manuscript.

R2C18: line 233. Change "increase" to "increases"
Reply: Will revise in the manuscript.

R2C19: line 270-271. Consider changing "due to the ignorance of" to "due to ignoring."

Reply: Will revise in the manuscript.

R2C20: line 285. Change semicolon to period.

Reply: Will revise in the manuscript.

R2C21: line 433. Change "e-f" to "d-f"

Reply: Sorry for the typo. Will revise in the manuscript.

R2C22: line 287. Add period.

Reply: Will revise in the manuscript.

References:

Liu et al., Global drought and severe drought-affected population in 1.5oC and 2oC warmer world, Earth System Dynamics, 9, 267-283, 2018

To Reviewer #3:

R3C1: This very innovative and important study shows that when the familiar Palmer Drought Severity Index (PDSI) is computed directly from global Earth System Model output of precipitation, evaporation, runoff and soil moisture storage (rather than box-modeling all those quantities from an offline-computed potential evaporation of questionable accuracy as is traditionally done), the dire projections of ubiquitous future global drought from those traditional studies vanish. Instead, the PDSI projections become both wetting and drying depending on region, consistent with the *direct* simulations of runoff, deep-layer soil moisture, etc. by the ESMs but not with the traditional PDSI studies.

This is a key methodological advance and shows that the PDSI index itself is not flawed under climate change, rather its known problems stem from the traditional potential evaporation input which is inaccurate, leading to inaccurate inferred water flux changes. The inaccuracy of the traditional potential-evaporation input is because leading-order biological effects of changing $CO_2$ and vapor pressure are taken into account in the ESMs but not in the potential-evaporation calculation, as the authors show well here.

I recommend only minor revisions before publication, since I was anonymous Reviewer #2 on the earlier version of this study that was originally submitted to Environ. Res. Lett., and I already had my concerns largely addressed during that review process at that journal. My strongly recommended minor revisions are listed below.

Reply: Thanks for your favorable evaluation of our study. Your individual comments are replied below.

R3C2: 30: This kind of parenthetical remark/qualification is appropriate for the body text, but I don't think is needed for the Abstract - it makes the Abstract too complicated and clunky. At least, that is how I read it. So I think you should either remove or greatly shorten this remark. You can put something like this in the body text instead.

Reply: We will remove this in the revised manuscript.

R3C3: 54-57, 94-96, 122, 226, 271, 418: Should also mention the impact of increasing/elevated vapor pressure deficit, as you do in the Abstract. The direct effect of $CO_2$ is only part of the story, as you explain well at 235-237 but the text does not reflect here at all.

Reply: Will do in the revised manuscript.

R3C4: 88-89: It should be clarified here that this corresponds to the center stream in Figure 1, parallel to how you point out the right stream and left stream later in the paragraph.
Reply: Will do in the revised manuscript.

R3C5: 117-120: Similarly, this should mention that it is the right stream in Fig 1.
Reply: Will do in the revised manuscript.

R3C6: 120-123: Similarly, this should mention that it is the left stream in Fig 1.
Reply: Will do in the revised manuscript.

R3C7: 126-135: Similarly, this should mention that it is the center stream in Fig 1.
Reply: Will do in the revised manuscript.

R3C8: 178: Should be 3d, not 3e.
Reply: Sorry for the typo. We will correct it in the revised manuscript.

R3C9: 238: As stated in previous review for Environ. Res. Lett., "our results" on this line will be read by most readers as meaning "the current study" (even though that's not what you actually mean.) Since it's actually Yang et al (2019) that showed this key fact (not the current study), this needs to be rephrased to make that clear. It's largely the same authors, but different study, and the distinction is important.
Reply: We will rephrase this sentence in the revised manuscript.

R3C10: Fig 2: Caption should point out which rows respectively correspond to the left, right and center streams of Fig 1.
Reply: Good suggestion; we will update the caption of Figure 2 in the revised manuscript.

R3C11: Fig 3d-f: Stippling when >50% of models agree on sign of change is trivial - this will almost always be true (unless exactly 8 models have increase and exactly 8 have decrease.) Rather, you should stipple when, say, >67% or >80% of models agree on sign of change. This better filters out changes that are just noise.

It is true that I suggested 50% threshold in previous review, but that was for models with dPDSI < -1, not for basic sign of change! 50% makes sense if the criterion is dPDSI < -1 because that's not likely to occur by chance. But it doesn't make sense for dPDSI < 0 or dPDSI > 0, since that occurs most of the time by chance (unless

*exactly* 8 models happen to have a decrease!)

Reply: We are sorry about this but there is a typo in the caption of Figure 3. The black dots actually show the same sign detected in at least 13 models (so >80%). We will correct it in the revised manuscript.

R3C12: Supp Fig S1: This is greatly appreciated, but I think it would have an even greater impact if you reversed the color scale in panels b and c (i.e. make negative green/blue, and positive yellow/brown.) This is because in this context we are thinking of E as a loss term in the water budget, and so increasing trend of E -> more drying. (I know that in other contexts/purposes more E -> wetter, but here the purpose is clearly to indicate that panel c is not as "drying" as panel b, so the colors should intuitively reflect that!)

Reply: Thanks for the suggestion. We will recolor the curves in the revised manuscript.

---

## Author Response (AR1)

**Response to Reviewers' comments**

We greatly appreciate the reviewers providing valuable and constructive comments on our manuscript HESS-2019-701. We seriously considered each comment and revised/improved the manuscript accordingly. The individual comments are replied below. In the following the reviewer comments are black font and our responses are blue and to assist with navigation we use codes, such as R1C2 (Reviewer 1 Comment 2).

To Anonymous Referee #1

R1C1: General comments: This paper analyzes projected changes in PDSI. It compares PDSI estimates obtained using potential evapotranspiration with and without accounting for the response of vegetation to increasing atmospheric CO2 concentration, as well as a direct estimate based on hydrological output from CMIP5 climate models. The main point is that there is no significant global drying trend based on PDSI, and the reason this was previously suggested is that offline impact models did not account for the response of vegetation to increased CO2. As noted by the authors (page 3, lines 54–57), several recent studies have already pointed out this problem when computing ET offline.

The valid point the authors make of refuting a general rule of "warming leads to drying" should not be interpreted as there will be no drying. The authors could try to make this even clearer by further emphasizing the projected increase in land area fraction under extreme conditions of water availability as well as the uncertainties in the projections.

Overall, the manuscript is well-structured and clearly conveys its main point. Nonetheless, it would be useful to further discuss some aspects of the methodology and address potential caveats of the PDSI.

Reply: Thanks for your encouraging and constructive comments. Your individual comments are replied below. We have changed the title and the text to avoid misleading the readers as "there will be no drying" and to focus on our key information that "we use direct climate model fluxes as inputs to PDSI and compare that with traditional PDSI. We find traditional PDSI overestimates projected drought. We also find that you can do a reasonable job using traditional PDSI but with $CO_2$ effects incorporated."

Specific comments

R1C2: Although PDSI has been a widely used index, it is not exempt from caveats.

When analyzing projected changes in drought (water availability) it would be beneficial to also directly show the changes in relevant variables like soil moisture and precipitation minus evapotranspiration. Although results for SPEI are presented in the supplement, a summary of trends in projected soil moisture anomalies would be a suitable complement to the manuscript. Particularly, maps of the trends would provide a more comprehensive picture as opposed to the global averages.

Reply: The maps of trends in soil moisture and precipitation minus evapotranspiration have been shown in a few previous publications (e.g., Berg et al., 2017; Greve et al., 2017; Swann et al., 2016; Yang et al., 2019); we have cited these papers and summarized/discussed their findings in the manuscript.

An important motivation of this study is actually based on these previous findings that total soil moisture (and root-zone soil moisture) does not show notable changes and precipitation minus evapotranspiration (or runoff) shows a slightly increase but estimated drought increases substantially in the coming century. The current study is designed to solve this contradiction. Several studies have pointed out the issue of ignoring the $CO_2$ effect in offline ET (and/or runoff) estimations (as noted by the reviewer), with the findings have important implications on drought changes. This study goes one step further by directly focusing on drought, using a widely used drought index – PDSI.

The spatial patterns of PDSI trend are shown in Figure 3. The global averaged PDSI series was intended to give an overall comparison between different PDSIs at the global scale (given comments by the editor and other reviewers, we removed this global average PDSI series from the main text in the revised manuscript).

R1C3: It appears that the climatically appropriate for existing conditions (CAFEC) coefficients are estimated for the entire period 1901–2100 (if this is the case, it should be explicitly stated). This seems counterintuitive to me when analyzing projected changes. Why would it not be more meaningful to estimate the soil moisture anomalies relative to some reference conditions from the past or present, e.g. 1901–1960 as for SPEI in Fig. S4?

Reply: Both PDSI and SPEI are calculated for the entire period 1901-2100 (both indices calculate the monthly departure from climatological means, and the climatological means are computed as the mean over 1901-2100). With the calculated SPEI series, in Fig. S4 (now supplementary Figure S3), we show the long-term SPEI change relative to the 1901-1960 mean to better highlight the changes.

In the revised manuscript, we have made this point clear in the method section (Line

116-117). In addition, we have removed the global average PDSI series (Figure 4 in the original submission) from the main text.

R1C4: It would be relevant to discuss and/or provide sensitivity tests to assumptions underlying the calculation of PDSI. For example, what value was selected for the available water capacity (AWC)? Is it constant in space? Are the values model dependent?

Reply: There is only one parameter (AWC) needed in PDSI calculation, and is derived from the Global Gridded Surfaces of Selected Soil Characteristics (https://webmap.ornl.gov/ogcdown/dataset.jsp?ds_id=569). We have described this in more detail in the revised manuscript (Line 117-119). The sensitivity of PDSI to AWC has been examined in a previous study (Sheffield et al., 2012), and the authors found that changes in AWC have only very minor impact on PDSI estimates. This is now mentioned in the revised manuscript (Line 119-120).

R1C5: It would be insightful to know more about the variability of PDSI given that all data is already available. For example, maps of change in the standard deviation of PDSI from a future period relative to present-day can be shown in the supplement.

Reply: Done. Revised as suggested (Supplementary Figure S4)

R1C6: The manuscript concludes (page 11, lines 273–274) highlighting the increased spatial variability in surface hydrological conditions. In this context, it could be appropriate to also discuss local changes in temporal variability, see Kumar et al. (2013). Kumar, S., Lawrence, D. M., Dirmeyer, P. A. & Sheffield, J. Less reliable water availability in the 21st century climate projections. Earth's Futur. 2, 152–160 (2013).

Reply: Done. Revised as suggested (Line 284), with the suggested paper now cited (thanks for the comment).

R1C7: Page 6, lines 129–132: Potential for discussion. Differences between total soil-depth representation in CMIP models may lead to systematic differences in PDSI estimates from individual models.

Reply: Done. We now mention this point in the revised manuscript (Line 137-139). In addition, we would like to point out although the absolute PDSI value might be different, differences in soil-depth are unlikely to affect the PDSI changes (as per Sheffield et al., 2012, see R1C4 above).

In addition, we updated our results by showing mean and range (estimates from all individual models) instead of standard deviation in the revised manuscript (Figure 5,

S1-S3 in the revised manuscript). This better shows the difference between individual models.

R1C8: It should be noted that the discussed response of vegetation to increasing CO2 applies to transpiration, but not to evaporation from the soil and canopy as well as snow sublimation. In this case, increasing CO2 and temperature would have a direct effect towards increasing evaporation.
Reply: We agree with this point and we add a sentence in the revised manuscript to explicitly state that increasing $CO_2$ only impacts vegetation transpiration (Line 55-58).
Relevant text read: "This vegetation-[$CO_2$] response only impacts transpiration, not soil evaporation, interception from vegetation surfaces or sublimation in snow environments, however it should be noted that transpiration dominates ($\sim 65\%$) global terrestrial evaporation (Lian et al., 2018; Zhang et al 2016)."

R1C9: Fig. 3 shows that even for direct CMIP5 output there can be a considerable increase in the land fraction experiencing extreme drought/moist conditions. These areas could be even larger if we were to consider the full spread of the CMIP5 ensemble as opposed to plus/minus one standard deviation. Is it reasonable to consider that differences in how individual models represent the response of vegetation to increasing CO2 could explain the spread in CMIP5 projections? This may be an important discussion point for the paper.
Reply: We have updated our results by showing mean and range (estimates from all individual models) instead of standard deviation in the revised manuscript to better show the difference between individual models (Figures 5, S1-S3). However, it is beyond the scope of this manuscript to investigate how individual CMIP5 models deal with the response of vegetation to increasing $CO_2$. To the best of our knowledge, most (if not all) CMIP5 models adopt the Farquhar photosynthesis model to estimate assimilation rate, which is then coupled with the Ball-Barry model to estimate stomatal resistance. So, the response of vegetation to increasing $CO_2$ in most of the CMIP5 models is essentially the Farquhar response. So, the difference among models is unlikely caused by how the response of vegetation to increasing $CO_2$ represented in the model but more likely caused by the simulated difference in the controlling environmental factors (e.g., temperature, water availability, radiation, etc.) that modify the Farquhar response.

R1C10: What is the reason why this particular subset of 16 CMIP5 models was used and not all models that are available?
Reply: These particular 16 CMIP5 models were used because these 16 models

provide all outputs we need, in particular runoff estimations. We have explicitly stated this in the revised manuscript (Line 85-86).

Relevant text read: "These 16 CMIP5 models were selected as they output all variables, including runoff, that are needed for the analysis performed herein."

R1C11: Trends in vegetation greening are mentioned in the abstract. The following reference about hidden vegetation browning could be helpful. Pan, N., Feng, X., Fu, B., Wang, S., Ji, F. and Pan, S.: Increasing global vegetation browning hidden in overall vegetation greening: Insights from time-varying trends, Remote Sens. Environ., 214, 59–72, doi:10.1016/J.RSE.2018.05.018, 2018.

Reply: We have changed the relevant text as "the overall global greening", which implies that there are also scattered browning trend. We highlight greening here as it is the very big picture. However, the main text is all about climate model projections (which show an even more consistent greening trend across the globe), so the observed browning trend is not relevant here.

Technical comments

R1C12: In Palmer (1965), equation 26 appears to use monthly recharge (R) instead of long-term average R. This might be worth double checking since it seemed to me the average is used in the provided scripts.

Reply: The equation 26 in Palmer (1965) does not use monthly R but monthly climatological R: long-term mean R for each month. We follow that.

That equation is to estimate monthly weighting factors, so each month has only one weighting factor.

R1C13: Lines 45 and 225: Inconsistency in the reference Lehner et al., 2017 or 2018? There is only one reference entry.

Reply: Apology for the typo. It is Lehner et al., 2017. We have corrected it in the revised manuscript.

R1C14: Page 10, lines 23: I would delete the word "also" since the effects are opposite.

Reply: Done. Revised as suggested (Line 241).

R1C15: Page 11, line 273: Is Fig. 3b–f correct? Or Fig. 3b–c?

Reply: Done. We have renumbered the figures and carefully checked the text to ensure they are correctly referenced in the text.

R1C16: Figure 3: The selection criteria for where to have the black dots does not seem optimal. As it is now, it is showing all regions where the mean and median of PDSI have the same sign. I would suggest a different threshold for model agreement, e.g. black dots where at least 2/3 of the models agree in sign. Alternatively, it could be useful to include in the supplement maps of model agreement that are complementary to Figs. 3d–f.

Reply: There is a typo in the caption of Figure 3. The black dots actually show the same sign detected in at least 13 models (so >80%). We have corrected it in the revised manuscript (see caption of Figure 3).

R1C17: Page 8, line 194 and 196: It should be Fig. 4.

Reply: Apology for the typo. Since we have removed the global average PDSI series from the main text in the revised manuscript, this sentence has been deleted too.

To Anonymous Referee #2:

R2C1: General Comments This report is a welcome contribution to the ongoing discussion in the literature regarding how to characterize changes in drought incidence under the changing climate. The paper is a follow-up to the paper by Yang et al (2019), which presents an equation that generally captures the variation of effective stomatal resistance within CMIP5 models as a function of atmospheric carbon dioxide concentration. In this paper, that relation is used to show how a popular drought index (the PDSI) can be adapted to characterize drought in our world of greenhouse warming. A readily available and simple offline alternative to the usual PDSI (and, in particular, to the Allen et al. form of the Penman-Monteith equation) will likely be of value to climate-change impacts analysts, many of whom may not be familiar enough with the biological processes in play or have the resources to model the processes with greater fidelity. That being said, it is important to evaluate the performance of the modified index carefully and to lay out clearly the assumptions and limitations in one place.

To some extent, the literature in this area has had a certain feel of X-vs-Y to it, X being increase of drought, and Y being no change in drought to speak of. This paper moves a bit toward the middle in acknowledging increases in drought incidence, but the overall presentation still has the feel of Y. Some specific suggestions for movement toward what might be a more balanced presentation are offered below for the authors' consideration.

Reply: Thanks for your encouraging and constructive comments. Your individual comments are replied below.

Specific Comments:

R2C2: The title "Little Change. . ." (which echoes that of Sheffield et al.) places the paper in the Y category mentioned in the General Comments above. To me, and perhaps to other readers, "little" implies something along the lines of "nothing to worry about." The authors might consider modifying the title to avoid that implication.

Reply: Thanks very much for the point. We have changed the title to: Comparing PDSI drought assessments using the traditional offline approach with direct climate model outputs.

R2C3: The reference to "PDSI" in the title, without qualification, is potentially confusing. Other publications (as well as this one) have shown that the usual PDSI equation applied to climate-change projections do imply increased drought. Would it be appropriate to change "PDSI" to "Co2-aware PDSI" or something else that

conveys that idea?

Reply: Please see our reply to R2C2.

R2C4: In general, the paper does a good job of citing the relevant literature. However, it's not clear to me that the abstract does justice to the previous literature (including the authors' own works) when it uses the phrase "resolve a paradox."

Reply: We have removed the phrase "resolve the paradox" in the revised manuscript.

R2C5: It's not immediately apparent what it means for the abstract to say that "global PDSI_CMIP5" remains generally unchanged. If this refers to the global average of the time average of the ensemble average of PDSI, then it is possible that the element of variability in space, in time, and across models could be lost in translation. It's hard to think about "drought" without considering variability.

Reply: We have revised the abstract to avoid any misunderstanding and/or misinterpretation of such. The new abstract reads:

"Anthropogenic warming has been projected to increase global drought for the 21st century when calculated using offline drought indices. However, this contradicts observations of the overall global greening and little systematic change in runoff over the past few decades and climate projections of future greening with slight increases in global runoff for the coming century. This calls into question the drought projections based on offline drought indices. Here we calculate a widely-used conventional drought index (i.e., the Palmer Drought Severity Index, PDSI) using direct outputs from 16 CMIP5 models (PDSI_CMIP5) such that the hydrologic consistency between PDSI_CMIP5 and CMIP5 models is maintained. We find that the PDSI_CMIP5-depicted drought increases (in terms of drought severity, frequency and extent) are much smaller than that reported when PDSI is calculated using the traditional offline approach that has been widely used in previous drought assessments under climate change. Further analyses indicate that the overestimation of PDSI drought increases reported previously using the traditional PDSI is primarily due to ignoring the vegetation response to elevated atmospheric $CO_2$ concentration ($[CO_2]$) in the offline calculations. Finally, we show that the overestimation of drought using the traditional PDSI approach can be minimized by accounting for the effect of $CO_2$ on evapotranspiration."

R2C6: The statement in the abstract that "projected increase in PDSI drought reported previously is primarily due to ignoring the vegetation response" seems somewhat overstated when I look at Figure 3, which suggests that the increase is about 50% or so due to ignoring the biological response, leaving another 50% that is not due to that.

Reply: We have revised the relevant text in the abstract to avoid any misunderstanding

and/or misinterpretation of such. In particular, in stead of saying "projected increase in PDSI drought reported previously", we now use "the overestimation of PDSI drought increases reported previously" (Line 31).

R2C7: I did not carefully evaluate what was implied by lines 138-140: "The PDSIs were calculated using outputs of each CMIP5 model in turn, and the ensemble PDSIs (averaging PDSIs over models) were used in the following analyses," but that passage gave me pause. Won't averaging across models reduce both the temporal and spatial variability and thereby impact drought estimates?
Reply: We removed the global average PDSI series (Figure 4 in original submission) from the main text and updated (the new) Figure 5 to show the range of drought/moist areas projected by all individual models. In addition, results in Figure 5 are not ensembles but the results agreed in at least 8 models (as suggested by reviewer #3, i.e., R3C11).

R2C8: lines 208-210. The criterion for substantial increase in drought appears to based on the change in the average value of PDSI rather than the change in the exceedance of a threshold. Is that a good measure?
Reply: The focus here was the increase in drought (decrease in PDSI by definition). A place changes from extreme moist to mild moist is also considered as an indication of potential drought increase. The same analysis has been applied in a few previous studies (e.g., Liu et al., 2018) so we use the same approach to be able to compare with results from others.
Changes of PDSI exceeding a certain threshold indicate changes area under drought, and these results are given in Figure 5. In addition, we add a figure showing trends in months with PDSI exceeding a certain threshold to show changes in time under drought (Figure 4).

R2C9: lines 210-212. It was stated earlier that ensemble averages were used for the analysis. It's quite possible I've read through the paper too quickly; the authors might consider taking precautions to avoid letting the casual reader get confused.
Reply: This is not the ensemble result (Figure 6 in the revised manuscript); instead, it is the result that shared by at least 8 of the 16 CMIP5 models (as suggested by reviewer #3, i.e., R3C11). This is now made clear in the revised manuscript (Line 217-219).

R2C10: lines 233-235. I am confused by "yet on the other hand" (which by the way sounds redundant in itself) combined with "also," since both effects are working in the same direction. The "also" seems out of place here. I get that "also" here was

meant in the sense of "and here's another thing it does," but the current sentence structure doesn't work for me.

Reply: We have deleted "also" in the revised manuscript (Line 241).

R2C11: line 242-247. I think this is another place where the authors could relax away from the "Y" position mentioned in the general comments. It seems to me that the dryness near the surface might be important for wildfire risk and perhaps for various biological processes that take place close to the surface. This idea might even be allowed to bubble up into the abstract.

Reply: Following your suggestion, we have extended the discussion on possible impacts of surface soil moisture decline in the revised manuscript (Line 255-264).

R2C12: Figures 1 and 2. The creation of these figure to convey what's going on is appreciated. Figure 2 takes a while to understand. It might help if the four black arrows and the plus signs were removed. It also might help if there were another column for how Ep is computed.

Reply: Done. Revised as suggested. The second column from the left (under Meteorological Inputs) shows how $E_P$ was computed for each PDSI.

R2C13: Figure 3. A map of trend in PDSI doesn't seem as useful as a map of trend in exceedance of some substantial value of PDSI.

Reply: The map of PDSI trend gives a general information on how PDSI changes. Per your comment we now also show the trend for the area under drought (defined by the PDSI exceeding a nominated threshold) (Figure 5). Following your suggestion, we have also added a map showing time (i.e., the number of months) in each year with a PDSI value exceeds a certain threshold (Figure 4).

R2C14: lines 435-436. "where the same sign of the PDSI trend is identified in at least 8 out of the 16 CMIP5 models.." Taken alone, without additional explanation in the caption, it seems like this would be true everywhere.

Reply: There is a typo in the caption of Figure 3. The black dots actually show the same sign detected in at least 13 models (so >80%). We have corrected this in the revised manuscript.

R2C15: Figure 4. This figure averages out a lot of information. Do the benefits of its inclusion outweigh the possible misunderstanding that it might generate? See also related comment above regarding "global PDSI_CMIP5" in abstract.

Reply: As also suggested by the editor, we removed this figure from the main text in the revised manuscript.

R2C16: Figure 5. As mentioned elsewhere, change in (expected value of) PDSI might not be the best metric for change in drought. Change in exceedance of thresholds might be better. I wouldn't be surprised if these were quite parallel, but to leave that taken for granted could weaken the overall impact of the paper.

Reply: Please see our reply to R2C8.

In addition, we understand this reviewer's concern of using changes in PDSI as the measure. However, using changes in exceedance of thresholds may also incur other issues. For example, if we chose PDSI = -1 as the threshold for drought, then locations having a PDSI = -2 in the baseline period and a PDIS = -1.5 in the future period (or PDSI = -1.5 in the baseline period and PDSI = -1.51 in the future period) will be identified as places with a substantial drought increase.

Technical Corrections

R2C17: line 93. Delete "and"

Reply: We have rewritten this part and this comment is not relevant in the revised manuscript.

R2C18: line 233. Change "increase" to "increases"

Reply: Done. Revised as suggested (Line 240).

R2C19: line 270-271. Consider changing "due to the ignorance of" to "due to ignoring."

Reply: Done. Revised as suggested (Line 279).

R2C20: line 285. Change semicolon to period.

Reply: Done. Revised as suggested (Line 271).

R2C21: line 433. Change "e-f" to "d-f"

Reply: The figures have been restructured and renumbers in the revised manuscript. We have carefully checked all figure captions to avoid such mistakes.

R2C22: line 287. Add period.

Reply: Done. Revised as suggested (Line 296).

References:

Liu et al., Global drought and severe drought-affected population in 1.5°C and 2°C warmer world, *Earth System Dynamics*, 9, 267-283, 2018

To Reviewer #3:

R3C1: This very innovative and important study shows that when the familiar Palmer Drought Severity Index (PDSI) is computed directly from global Earth System Model output of precipitation, evaporation, runoff and soil moisture storage (rather than box-modeling all those quantities from an offline-computed potential evaporation of questionable accuracy as is traditionally done), the dire projections of ubiquitous future global drought from those traditional studies vanish. Instead, the PDSI projections become both wetting and drying depending on region, consistent with the *direct* simulations of runoff, deep-layer soil moisture, etc. by the ESMs but not with the traditional PDSI studies.

This is a key methodological advance and shows that the PDSI index itself is not flawed under climate change, rather its known problems stem from the traditional potential evaporation input which is inaccurate, leading to inaccurate inferred water flux changes. The inaccuracy of the traditional potential-evaporation input is because leading-order biological effects of changing $CO_2$ and vapor pressure are taken into account in the ESMs but not in the potential-evaporation calculation, as the authors show well here.

I recommend only minor revisions before publication, since I was anonymous Reviewer #2 on the earlier version of this study that was originally submitted to Environ. Res. Lett., and I already had my concerns largely addressed during that review process at that journal. My strongly recommended minor revisions are listed below.

Reply: Thanks for your favorable evaluation of our study. Your individual comments are replied below.

R3C2: 30: This kind of parenthetical remark/qualification is appropriate for the body text, but I don't think is needed for the Abstract - it makes the Abstract too complicated and clunky. At least, that is how I read it. So I think you should either remove or greatly shorten this remark. You can put something like this in the body text instead.

Reply: Done. We have removed this additional remark in the revised manuscript.

R3C3: 54-57, 94-96, 122, 226, 271, 418: Should also mention the impact of increasing/elevated vapor pressure deficit, as you do in the Abstract. The direct effect of $CO_2$ is only part of the story, as you explain well at 235-237 but the text does not reflect here at all.

Reply: We have mentioned the VPD impact in Line 56 as you suggested. However, as

we have carefully gone through other places you mentioned, we do not think they are biased statements. We use the term "CO$_2$ effects on vegetation" as a lumped impact including both direct and indirect pathways. Since we have explained them in the introduction and discussion, we prefer to keep using the "CO$_2$ effects on vegetation" as the overall impact to improve the flow of the manuscript.

R3C4: 88-89: It should be clarified here that this corresponds to the center stream in Figure 1, parallel to how you point out the right stream and left stream later in the paragraph.
Reply: Done. Revised as suggested (line 92). Noting that we use the word 'column' (or 'approach') rather than 'stream' to avoid any potential confusion for readers who are better reading languages other than English.

R3C5: 117-120: Similarly, this should mention that it is the right stream in Fig 1.
Reply: Done. Revised as suggested (line 125).

R3C6: 120-123: Similarly, this should mention that it is the left stream in Fig 1.
Reply: Done. Revised as suggested (line 130).

R3C7: 126-135: Similarly, this should mention that it is the center stream in Fig 1.
Reply: Done. Revised as suggested (line 133); and thanks for these (i.e., R3C4 to R3C7, inclusive) really good comments that mean Figure 1 becomes a 'backbone' for our analysis.

R3C8: 178: Should be 3d, not 3e.
Reply: Sorry for the typo. We have corrected in the revised manuscript (Line 180-183).

R3C9: 238: As stated in previous review for Environ. Res. Lett., "our results" on this line will be read by most readers as meaning "the current study" (even though that's not what you actually mean.) Since it's actually Yang et al (2019) that showed this key fact (not the current study), this needs to be rephrased to make that clear. It's largely the same authors, but different study, and the distinction is important.
Reply: Done. Revised as suggested (Line 247); thanks for the comment.

R3C10: Fig 2: Caption should point out which rows respectively correspond to the left, right and center streams of Fig 1.
Reply: Good suggestion; we have updated the caption of Figure 2 following your suggestion in the revised manuscript, noting (as per R3C4) we use the word 'column'

rather than 'streams'.

R3C11: Fig 3d-f: Stippling when >50% of models agree on sign of change is trivial - this will almost always be true (unless exactly 8 models have increase and exactly 8 have decrease.) Rather, you should stipple when, say, >67% or >80% of models agree on sign of change. This better filters out changes that are just noise.

It is true that I suggested 50% threshold in previous review, but that was for models with dPDSI < -1, not for basic sign of change! 50% makes sense if the criterion is dPDSI < -1 because that's not likely to occur by chance. But it doesn't make sense for dPDSI < 0 or dPDSI > 0, since that occurs most of the time by chance (unless *exactly* 8 models happen to have a decrease!)

Reply: We are sorry about this but there is a typo in the caption of original Figure 3. The black dots actually show the same sign detected in at least 13 models (so >80%). We have corrected this in the revised manuscript; again we are sorry for the inconvenience.

R3C12: Supp Fig S1: This is greatly appreciated, but I think it would have an even greater impact if you reversed the color scale in panels b and c (i.e. make negative green/blue, and positive yellow/brown.) This is because in this context we are thinking of E as a loss term in the water budget, and so increasing trend of E -> more drying. (I know that in other contexts/purposes more E -> wetter, but here the purpose is clearly to indicate that panel c is not as "drying" as panel b, so the colors should intuitively reflect that!)

Reply: Done. Revised as suggested; and again thanks.

[revised manuscript text omitted]

---

## Referee Report (RR1)

**Review of "Comparing PDSI drought assessments using the traditional offline approach with direct climate model outputs"**

**General comment**

The authors have thoroughly answered the points raised during the previous review round, and incorporated changes that improved the manuscript.

**Specific comments**

1. Page 3, line 58: Perhaps it is relevant to note the variability across CMIP5 models of the ratio of transpiration over evapotranspiration, as well as its mean underestimation compared to the observational estimate provided in the manuscript.

2. Page 8, line 199: Use *least* instead of *fewest*.

3. Figure 6d: Double check. Visually, the areas in Figures 6e and 6f appear to be larger than indicated by the bar plot in Figure 6d. Also, it appears to be different than the values indicated in page 9, line 227.

---

## Author Response (AR2)

We greatly appreciate the editor and reviewers again providing valuable and constructive comments on our manuscript. We seriously considered each comment and revised the manuscript accordingly. The individual comments are addressed in the following response letter and the manuscript has been revised to accommodate the changes (changes are marked in a different color). In the following the reviewer comments are black font and our responses are blue and to assist with navigation we use codes, such as R3C2 (Reviewer 3 Comment 2). The line numbers correspond to line numbers in the clean version of the revised manuscript.

To Reviewer #1

R1C1: General comment: The authors have thoroughly answered the points raised during the previous review round, and incorporated changes that improved the manuscript.
Reply: Reply: Thanks for the assessment and following suggestions.

R1C2: Page 3, line 58: Perhaps it is relevant to note the variability across CMIP5 models of the ratio of transpiration over evapotranspiration, as well as its mean underestimation compared to the observational estimate provided in the manuscript.
Reply: Done. This information and relevant reference are given in the revised manuscript (Line 63-64)

R1C3: Page 8, line 199: Use least instead of fewest.
Reply: Done. Revised as suggested (Line 209).

R1C4: Figure 6d: Figure 6d: Double check. Visually, the areas in Figures 6e and 6f appear to be larger than indicated by the bar plot in Figure 6d. Also, it appears to be different than the values indicated in page 9, line 227
Reply: Oops! We changed the color of Figure 6a and 6d in the last revision (as per reviewer #3) and have no idea why we did such a mistake. We have now corrected the bars in Figure 6a and 6d. Thanks very much for this sharp pick-up.

To Reviewer #3

R3C1: This revision addressed my comments almost entirely. I recommend publication pending the following minor revisions. The first suggested revision is an aspect I overlooked last review, but the other suggested revisions are simply follow-ons to changes the authors made in response to previous round of comments (mine and others'.) All should be very quick to implement.
Reply: Thanks for the positive assessment and the additional suggestions.

R3C2: 22-24: This is an important thought but is never fleshed out in the body text with full citations, though aspects of it are mentioned. I think it's worth including a paragraph or a few sentences in the body (specifically, the introduction) about this contradiction, so the reader knows where in the published literature to verify each of the claims you are stating (i.e. that greening occurred over the last few decades, that runoff had little systematic change in the last few decades, that slight runoff increases are projected for future, and that large greening is projected for future.) Otherwise the abstract has un-cited statements. In general, this contradiction is a powerful motivation for the rest of the paper, but almost seems to be forgotten about in the body text (except for the specific aspect about future runoff projections) after a very strong opening here in the abstract.
Reply: Done; very good comment. Revised as suggested (Line 45-49).

Line 45-49 reads: However, this substantial increase in projected drought contradicts observations of global vegetation greening and little systematic change in runoff over the past few decades and climate projections of future greening with slight increases in global runoff for the coming century (Alkama et al., 2013; Greve et al., 2017; Labat et al., 2014; Roderick et al., 2015; Milly and Dunne et al., 2016; Scheff et al., 2017; Yang et al., 2018; Yang et al., 2019; Zhu et al., 2016).

R3C3: 66: more specifically, this should say "right-hand column of Fig 1" or similar. In general, I would check through the whole paper and make sure that each time Fig 1 is called out, it mentions the appropriate column(s).
Reply: Done. Thanks for the suggestion (Line 72-73) and throughout the text.

R3C4: 194: The increases may "diminish", but they are still quite noticeable and widespread. Most interesting is that even when using PDSI_CMIP5 and PDSI_PM[CO2] (4b-c), there are *still* far more land areas with increases (yellow, orange, red) than with decreases (blue shades.) This should be pointed out explicitly. I.e. it should be pointed out that moving to PDSI_CMIP5 or PDSI_PM[CO2] reduces the magnitude of the drought frequency increase, but apparently does not reduce the very *widespread* distribution of drought frequency increase compared to drought frequency decrease.
Reply: Done. Revised as suggested (Line 202-205).

Line 202-205 reads: Yet, moving to PDSI_CMIP5 and PDSI_PM[$CO_2$] apparently do

not reduce the widespread distribution of drought frequency increase compared to drought frequency decrease (Figure 4b and 4c, i.e., there are more land areas with increasing drought frequency than with decreasing drought frequency).

R3C5: Fig 4: I'm not quite sure how to interpret the trend units (month yr-1) ? Shouldn't it be (month yr-2) , i.e., the trend-per-year in the number of months-per-year that exceed the given threshold? (Analogous to Fig S1?) Or do I misunderstand what you mean. In general, all the trends in the figures would be easier to understand if they were given per *century* of trend, rather than per year or per decade. I.e. multiply Fig 3 by 100 and change unit to PDSI/century, multiply Fig 4 by 100 and change unit to (month/year)/century (I think?), multiply Fig S1b,c by 100 and change unit to (mm/year)/century, multiply Fig S4 by 10 and change unit to PDSI/century. This is optional but I think helps readers a lot to understand the magnitude.
Reply: Done. The units are changed as suggested (Figs. 3, 4, S1, S4).

[revised manuscript text omitted]